# Training Neural Networks on Data Sources with Unknown Reliability

## Abstract

When data is generated by multiple sources, conventional training methods update models assuming equal reliability for each source and do not consider their individual data quality during training. However, in many applications, sources have varied levels of reliability that can have negative effects on the performance of a neural network. A key issue is that often the quality of data for individual sources is not known during training. Focusing on supervised learning, we aim to train neural networks on each data source for a number of steps proportional to the source's estimated relative reliability, by using a dynamic weighting. This way, we allow training on all sources during the warm-up, and reduce learning on less reliable sources during the final training stages, when it has been shown models overfit to noise. We show through diverse experiments, this can significantly improve model performance when trained on mixtures of reliable and unreliable data sources, and maintain performance when models are trained on reliable sources only.

## 1 Introduction

Data sources can have differing levels of noise but in many applications, are merged together to form a single dataset. In healthcare, for example, data sources (such as different devices or sites) are often combined together (Tomar & Agarwal, 2013; Baro et al., 2015), but may not provide the same level of data quality and could contain noisy features, incorrect labelling, or missing values. These problems, if unaddressed, can have detrimental effects on the performance and robustness of machine learning models (Zhang et al., 2021; Arpit et al., 2017; Neyshabur et al., 2017).

In this work, we consider the supervised learning case, in which a neural network is trained on data from multiple sources that are (ideally) producing unique features and labels from the same distribution, but where some sources are producing noisy features or labels at an unknown rate[1].

Two possible solutions for this context naturally arise: Preprocessing can be used to remove out-of-distribution observations from training (Gamberger et al., 2000; Thongkam et al., 2008; Delany et al., 2012) but, this requires the user to define "out-of-distribution" for the features and labels, and eliminating data assumes nothing can be learnt from noisy examples (Wang et al., 2018). Secondly, techniques for training neural networks on noisy data can be applied, of which many exist (Han et al., 2020; Song et al., 2022). In this case however, we are not utilising the information gained from knowing an observation's data source – which we show is useful, especially within settings of considerable noise.

We therefore propose Loss Adapted Plasticity (LAP); inspired by ideas from the literature, and loss tempering, it is a general method for training neural networks on multi-source data with mixed reliability. As we do not know the sources' true noise level a priori, we maintain a history of the empirical risk on data from each source that is used to temper the likelihood during training. This is done such that the number of steps a model is trained on a source is proportional to its estimated reliability. Hence, a model trained with LAP will benefit from seeing examples from all sources during early training, and reduce learning on less reliable data later in training, when the model is prone to memorising noisy data points (Zhang et al., 2021; Arpit et al., 2017; Liu et al., 2020; Xia et al., 2021). To illustrate our method's effectiveness, we present results on diverse settings and

---

[1]Training with missing data using our method could be achieved by randomly generating features or labels in place of missing values, but this is not a focus of our work.

datasets, in which we implemented or adapted baseline code from Han et al. (2018); Wang et al. (2021); Li et al. (2021) and Li et al. (2022). We find: (1) The proposed method, LAP, improves the performance of a neural network trained on data generated by sources with mixed reliability, and (2) maintains performance when no unreliability is present. We also show that (3) LAP is applicable to a variety of contexts and different noise types, and (4) we provide implementation details and code for further development and use (see Appendix A.1).

## 2 BACKGROUND

Noisy labels can be introduced at any point during collection and when human experts are involved, are practically inevitable (Song et al., 2022; Frenay & Verleysen, 2014; McNicol, 2004; Frenay & Verleysen, 2014): An expert might have insufficient information (Hickey, 1996; Dawid & Skene, 1979); expert labels are incorrect (Hickey, 1996); the labelling is subjective (Marcus et al., 1993); or there are communication problems (Zhu & Wu, 2004). Further, noisy features can be introduced through data processing, or faulty measurements (Li et al., 2021). With each of these issues affecting data sources differently, we can observe varied data quality.

The majority of the work on noisy data focuses on noisy *labels* where solutions usually target a combination of four aspects (Han et al., 2020; Song et al., 2022): sample selection, model architecture, regularisation, or training loss. IDPA (Wang et al., 2021) explores instance dependent noise, where label noise depends on an observation's features, in which they estimate the true label of confusing instances using model confidence during training. Han et al. (2018) propose "Co-teaching", a sample selection method, in which two neural networks are trained simultaneously. Data for which one model achieves a low training loss is selected to "teach" the other network, as they are assumed to be more reliable. This work exploits the fact that neural networks learn clean data patterns and filter noisy instances in early training (Zhang et al., 2021; Arpit et al., 2017), a fact that our work will take advantage of. Noisy *inputs* are additionally studied in Li et al. (2021), where they present RRL, employing two contrastive losses and a noise cleaning method using model confidence during training. This approach requires modifications to the model architecture and a $k$ nearest neighbours search at each epoch. However, these methods do not consider multi-source data.

In federated learning, a server trains a global model using local updates on each source (Konečný et al., 2016), but when sources are noisy, these algorithms often fail (Li et al., 2020). To tackle varied data quality, Li et al. (2022) propose ARFL, which learns global and local weight updates simultaneously. The weighted sum of empirical risk of clients' loss is minimised, where the weights are distributed by comparing each client's empirical risk to the average empirical risk of the best $k$ clients. The contribution to weight updates from the clients with higher losses are minimised with respect to the updates from other clients – forcing the global model to learn more from clients that achieve lower losses.

Further, Murphy (2022) discusses sensor fusion, where multiple observations from sensors with differing (*but known*) noise are taken, with the goal to calculate the true values. This is connected to our setting, but in our case, sources generate multiple unique observations, and source reliability is unknown.

Therefore, we are exploring a context between many; federated learning, sensor fusion, and learning with noisy data, but differing enough such that methods do not apply or under-perform compared to our source-aware solution. Nevertheless, Co-teaching, IDPA, RRL, and ARFL will form our baselines.

### MOTIVATION FOR NEW METHODS

We observe that work has so far focused on the problem of noisy data without considering individual sources. To see why this could be helpful, consider 10 data sources $\{s_i\}_{i=1}^{10}$ where one source, $s_c$, is producing noisy data with a probability of $0.5$ and all other sources are producing clean data. Given a new observation from the noisy source, $x^c \in s_c$, consider the probability that $x^c$ is *noisy*: $p(x^c \notin \mathrm{R})$, where $\mathrm{R}$ is the set of reliable data and assume we know the noise rate of $s_c$: $p(x \notin \mathrm{R} \mid x \in s_c) = 0.5$.

*Without knowing the data point's source:* The probability that a data point $x$ is unreliable (since $p(x \notin \mathrm{R} \mid x \notin s_c) = 0$) is: $p(x \notin \mathrm{R}) = p(x \in s_c)p(x \notin \mathrm{R} \mid x \in s_c) = 0.05$.

*When knowing the data point's source:* The probability that a data point $x^c$ is unreliable ($x^c \in s_c$) is $p(x^c \notin \mathrm{R} \mid x^c \in s_c) = 0.5$. Similarly, $p(x^{s \neq c} \notin \mathrm{R} \mid x^{s \neq c} \notin s_c) = 0$.

Clearly, the source value can provide information about noise that is helpful during model training.

## 3 METHODS

Inspired by Co-teaching, ARFL, and loss tempering, we propose Loss Adapted Plasticity (LAP): Briefly, we update a source reliability score as a function of the historical training empirical risk, which is used to re-weight loss such that the more reliable a source is, the longer the model trains on data from that source. Importantly, we use the fact that when training, neural networks learn non-noisy patterns before noisy data (Zhang et al., 2021; Arpit et al., 2017; Han et al., 2018; Arazo et al., 2019; Yu et al., 2019; Shen & Sanghavi, 2019) and are observed fitting to noisy examples only after substantial progress is made in fitting to clean examples (Arazo et al., 2019). Therefore, in early training the empirical risk on clean data is lower than noisy data (Appendix A.2).

TEMPERED LIKELIHOOD

In our setting, we have a dataset $\mathcal{D}$ collected from sources in $S$ with $\mathcal{D} = \bigcup_{s \in S} \mathcal{D}_s$ and $s$ as either a noisy source $s \notin \mathrm{R}$ or a non-noisy source $s \in \mathrm{R}$, where R represents the reliable sources. An optimal model on non-noisy data $\mathcal{D}_R$ (parameterised by $\theta$) would have log-likelihood: $\max_\theta \log p(\mathcal{D}_R|\theta)$. However, since $R$ is unknown during training, we propose to use the tempered log-likelihood on all data $\mathcal{D}$. With $f(\cdot) : [0, +\infty) \to [1, +\infty)$ as a positive and monotonically increasing function:

$$\log p(\mathcal{D}_R|\theta) = \sum_{s \in R} \log p(\mathcal{D}_s|\theta) \approx \sum_{s \in S} \frac{1}{f(C_s)} \log p(\mathcal{D}_s|\theta) = \log p_{\text{temp}}(\mathcal{D}|\theta) \tag{1}$$

Note the change of sum over $s \in R$ to $s \in S$. Here, $C_s$ is the number of training steps that we believe source $s$ contains noise harmful to training, minus the number of steps it is considered non-noisy (clipped at 0). Therefore during late training, $C_s$ is large for noisy sources and $C_s = 0$ for non-noisy sources, giving: a large temperature $f(C_s) >> 1$ for $s \notin \mathrm{R}$, and a low temperature $f(C_s) = 1$ for $s \in \mathrm{R}$, providing an approximation of $\log p(\mathcal{D}_R|\theta)$.

SOURCE RELIABILITY ESTIMATION

To calculate $C_s$, we use the assumption (discussed above) that neural networks achieve a lower empirical risk on clean data than noisy data in early training. Given a source $s$ for which we wish to update $C_s$, with all other sources as $s'$; using $p_\theta(\cdot) = p(\cdot|\theta)$; and setting $\lambda > 0$, we perform:

$$C_s = \begin{cases} C_s + 1, & \log p_\theta(D_s) < \mathtt{exptvar}_{s'}(\log p_{\text{temp},\theta}(D_{s'}), \lambda) \\ C_s - 1, & \text{otherwise} \end{cases}$$
$$C_s = \max\{0, C_s\} \tag{2}$$
$$\mathtt{exptvar}_\Delta(\blacktriangle, \lambda) = \mathbb{E}_\Delta[\blacktriangle] - \lambda \sqrt{\mathrm{Var}_\Delta[\blacktriangle]}$$

Intuitively, we increase $C_s$ and therefore the source's temperature, if its log-likelihood is at most $\lambda$ standard deviations less than the expected tempered log-likelihood on all other sources. Consequently, if no sources are noisy, and the distribution of negative log-likelihoods (NLLs) from the data sources forms a normal distribution, then we incorrectly increase the temperature of a non-noisy source $s$ with probability $p_\theta(\hat{L}_s \geq \lambda)$ where $L_s = -\log p_\theta(\mathcal{D}_s)$ is the NLL on data from source $s$, and $\hat{L}_s$ refers to the standardisation of the NLL using the mean and variance of the NLLs of all other sources, $L_{s'}$. When we have noisy sources, we expect that their temperature is large, $f(C_s) >> 1$, after a number of steps that is representative of their relative noise level – naturally enabling learning from noisy sources for a number of steps that reflects their "usefulness".

IMPLEMENTATION DETAILS

In the following, we present the implementation of our method as gradient scaling for clarity. Given a dataset $\mathcal{D}$ of features and labels that is generated by $S$ sources, $\mathcal{S} = \{s_1, ..., s_S\}$. We denote a subset of $\mathcal{D}$ generated by source $s$ as $\mathcal{D}_s \subset \mathcal{D}$, with each data point corresponding to a single source.

On a single update step of a model: we have the perceived *unreliability* for each source $C = \{C_s\}_1^S$ (initially, all sources are considered reliable and so $C_s = 0 \; \forall s$); a batch of features, labels, and source values from $\mathcal{D}$; and a history of training losses, $L$ with length $H$ for all sources. Here, $L \in \mathbb{R}^{S \times H}$, and $l_H^s \in L$ denotes the mean loss of data from source $s$ over the most recent batch containing data in $\mathcal{D}_s$ (hence, subscript $H$). A larger $C_s$ denotes a larger estimated noise level for source $s$. Each step, we update the value $C_s$ (and hence, temperature) for a source $s$ using the empirical risk of all other sources $s'$ as in Algorithm 1. Further, we define $f(C_s)$ from Equation 1:

$$1/f(C_s) = (1 - d_s) = 1 - \tanh^2(0.005 \cdot \delta \cdot C_s)$$

This choice of $f$ has some nice properties discussed in this section and in Appendix A.3. We refer to $d_s$ as the depression value; and $\delta$ is the depression strength, controlling the rate of depression.

---

**Algorithm 1** Calculating $C_s$ at a given step

---

**Require:** $\lambda > 0$ : Leniency
**Require:** $L \in \mathbb{R}^{S \times H}$ : Source loss history of length $H$
**Require:** $C \in \mathbb{R}^S$ : The current unreliability
   $L_s = L[s]$ {Loss history on source $s$}
   $C_s = C[s]$ {Unreliabilty for source $s$}
   $\mu_s = \texttt{mean}(L_s)$
   $W[s'] = 1/f(C[s'])$
   $\mu_{s'} = \texttt{weighted\_mean}(\, L[s'], \, \texttt{weights} = W[s'] \,)$
   $\sigma_{s'}^2 = \texttt{weighted\_var}(\, L[s'], \, \texttt{weights} = W[s'] \,)$
   **if** $\mu_s > \mu_{s'} + \lambda\sigma_{s'}$ **then**
      $C_s = C_s + 1$
   **else if** $\mu_s \leq \mu_{s'} + \lambda\sigma_{s'}$ **then**
      $C_s = \max\{\, 0, \, C_s - 1 \,\}$
   **end if**

---

During training, gradient contributions from source $s$ are then scaled as follows:

$$\hat{g}_s = (1 - d_s)g_s, \;\; d_s = \tanh^2(0.005 \cdot \delta \cdot C_s) \tag{3}$$

Where $\hat{g}_s$ is the gradient update contribution from a source $s$. As $(1 - d_s)$ is a scalar, this method can be interpreted as scaling gradient contributions (presented here), loss re-weighting as in Figure 1, or likelihood tempering as discussed previously in this section. Additionally, a model evidence interpretation of LAP can be found in Appendix A.4.

We use 0.005 to scale the depression strength $\delta$ for easier interpretation, whilst the use of $\tanh^2$ ensures that the scaling applied to $g_s$ is in $(0, 1]$ and small perturbations of $C_s$ around 0 do not have a significant effect on $\hat{g}_s$, making it more robust to randomness. For more detail, see Appendix A.3.

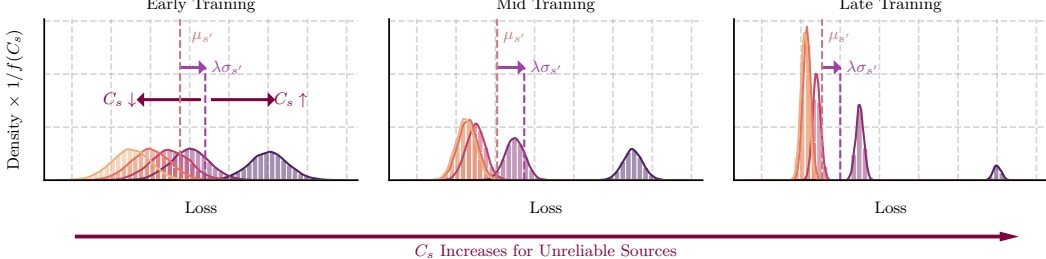

Figure 1: **Visualisation of Equation 2.** Each colour represents the loss values from a single source over a small number of steps, with its density weighted by its temperature, $w(s) = 1/f(C_s)$. This shows how sources contribute to $\mu_{s'}$ and $\sigma_{s'}^2$ as their $C_s$ changes during training and given the leniency $\lambda$. These values are synthetic and for demonstration.

INTUITIONS

Figure 1 shows how the training process evolves over time when using LAP. At first, all sources are considered equally reliable and so the weighted mean of the loss values is the mean of all sources. As

the source temperature changes, the weighed mean plus $\lambda$ standard deviations of the losses moves towards the sources with lower expected losses, allowing for more learning from the these over time.

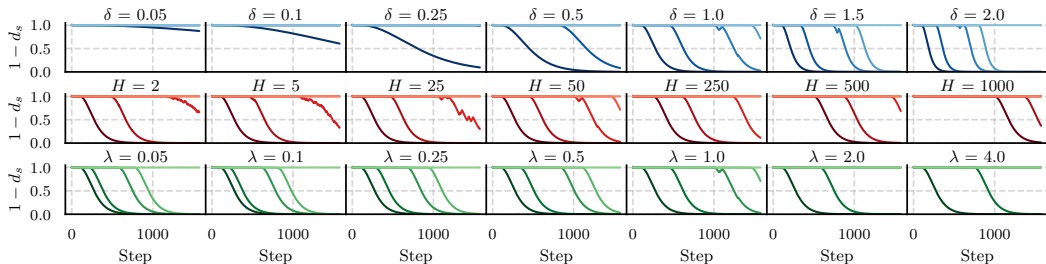

Figure 2: **Effect of the introduced parameters on training.** Section 3, introduces three parameters that control the effects of LAP. $1 - d_s$ is multiplied by the gradient (equivalently, loss) contribution from a given source before the model is updated. Here, we show these values for each source (the different coloured lines) during model training on synthetic data (Appendix A.5). Unless stated in the title of a given plot, the parameters of LAP were set to $H = 25$, $\delta = 1.0$, $\lambda = 1.0$. We had 5 sources with noise levels of 0.0, 0.025, 0.05, 0.25, and 1.0 (a darker colour indicates a higher noise rate).

Figure 2 illustrates how $1 - d_s$ (reciprocal of the temperature) in Equation 3 develops during training for each source and different $H$, $\delta$, and $\lambda$ values. The first row shows how varying the depression strength, $\delta$ affects the rate at which the learning on a source is reduced. The second row shows how the history length, $H$ influences $1 - d_s$, where we can see that a larger value allows for a "smoother" transition. With the exception of small or large values, this parameter has the least effect on training. The third row illustrates how the leniency, $\lambda$ in Equation 2 and Algorithm 1 affects the training. Small values of $\lambda$ reduce noisy sources' impacts influence earlier, whilst larger values miss some noisy sources all together. Here, the sources are coloured by their noise level, demonstrating that LAP reduced a source's influence on training in a number of steps corresponding to a sources noise level.

In our experiments, LAP parameters were set at $H = 25$, $\delta = 1.0$, $\lambda = 0.8$ unless otherwise stated in Appendix A.7. These were chosen based on the synthetic experiments presented in Figure 2 and the validation results on CIFAR-10 (Appendix A.7) as well as the detailed discussion of the effects of these hyperparameters provided in Appendix A.8.

## 4    EXPERIMENTS

### BASELINE METHODS

To contextualise our approach, we evaluate varied baseline methods: (1) ARFL, designed to tackle label and input noise in a federated learning setting (Li et al., 2022); (2) IDPA, a probabilistic method for instance dependent label noise (Wang et al., 2021) which modifies the training loss; (3) "Co-teaching", which trains two models simultaneously to perform sample selection based on loss values during training (Han et al., 2018); (4) RRL, which uses contrastive learning and a $k$ nearest neighbours search to enforce a smoothness constraint on learnt representations (Li et al., 2021), modifying model architecture and training loss; and (4) an identical model but without any specific modifications for tackling noise. Baseline methods were selected based on the availability of code, their use as baselines in the literature, and for variety in the methods used to evaluate the performance of LAP. Although our setting assumes noise is independent of features, we felt it was still beneficial to include IDPA, which is designed for instance dependent label noise. Note, that because ARFL is a federated learning approach, data cannot be shuffled in the same way as the other models, since each client trains on a single source.

It is important to note that both IDPA and Co-teaching require the training of a model twice, making LAP significantly less computationally expensive.

All baselines were implemented using the available code and trained using the recommended parameters with the model and data we test. For further details, see Appendix A.7.

EXPERIMENTAL DESIGN

To evaluate LAP, we employ various techniques to produce noisy data, extending the methods in Li et al. (2022); Wang et al. (2021); Han et al. (2018); Li et al. (2021) and test on eleven datasets from computer vision, healthcare time-series, natural language processing, and tabular regression for diverse experiments.

Following Li et al. (2022); Wang et al. (2021); Han et al. (2018); Li et al. (2021), we test our proposed method and baselines on CIFAR-10, CIFAR-100 (Krizhevsky, 2009), and F-MNIST (Xiao et al., 2017) forming many comparisons with the literature. Along with Tiny-Imagenet and Imagenet (Deng et al., 2009), these five datasets form well-studied computer vision tasks, easing reproducibility. Additionally, we use a human labelled version of CIFAR-10, titled CIFAR-10N (Wei et al., 2022), for which we use the "worst labels" allowing us flexibility in our experiments. We then study an electrocardiograph (ECG) dataset, PTB-XL (Wagner et al., 2020); a time-series classification task with the goal to classify normal and abnormal cardiac rhythms and for which noise can be simulated following Wong et al. (2012) and random labelling to understand LAP applied to time-series data with multiple noise types. Additionally, a sentiment analysis natural language task allows us to compare LAP against the literature, for which we employ the IMDB dataset (Maas et al., 2011) containing movie reviews and their sentiment. Next, we use the GoEmotions dataset (Demszky et al., 2020), a natural language emotion prediction task, which contains real-world imbalanced source sizes and class distributions, allowing us to test the robustness of LAP to varied source constructions. Finally, to illustrate our method's use for regression, we present results on the California Housing dataset (Pace & Barry, 1997). For further information, see Appendix A.6 and A.13.

To simulate data sources for CIFAR-10, CIFAR-100, F-MNIST, Imagenet, IMDB, and California Housing, data is uniformly split into 10 distinct groups, and for Tiny-Imagenet we use 100 groups to study larger numbers of sources. For CIFAR-10 and CIFAR-100, 4 and 2 of these sources are chosen to be noisy respectively; for F-MNIST, Imagenet, and Tiny-Imagenet 6, 5 and 40 are chosen; and for IMDB and California Housing 4 are chosen. These are in line with the noise rates used in the literature (often 20%, 40%, 50%). For CIFAR-10, in Appendix A.12 we also significantly increase the number of sources. To generate noise for the vision datasets, we extended the methods in Li et al. (2022): (1) Original Data: No noise is applied to the data; (2) Chunk Shuffle: Split features into distinct chunks and shuffle. This is only done on the first and/or second axis of a given input; (3) Random Label: Randomly assign a new label from the same code; (4) Batch Label Shuffle: For a given batch of features, randomly shuffle the labels; (5) Batch Label Flip: For a given batch of data, assign all features in this batch a label randomly chosen from the same batch; (6) Added Noise: Add Gaussian noise to the features, with mean $= 0$ and standard deviation $= 1$; (7) Replace with Noise: Replace features with Gaussian noise, with mean $= 0$ and standard deviation $= 1$.

For IMDB, GoEmotions, and California Housing we use random labelling.

PTB-XL was labelled by 12 nurses, naturally forming data sources. However, since this data is high quality, synthetic noise is required. We add Gaussian noise to sources' ECG recordings (simulating electromagnetic interference as in Wong et al. (2012)) and label flipping to simulate human error in labelling. This also allows us to test the setting with multiple noise types. Here, data from sources are upsampled so that each source contains the same number of observations. For experiments with PTB-XL, we linearly increase the number of noisy sources from 1 to 8 (out of 12 in total), and for each number of noisy sources we set the noise level for each source linearly from 0.25 to 1.0. For example, when training with 4 noisy sources, sources haves noise levels of 0.25, 0.50, 0.75, and 1.0.

Since GoEmotions is annotated by 82 raters, their indentification number is used to form the data sources. Some raters contributed a handful of data points, whilst others labelled thousands; with each rater providing a different distribution of class labels. This allowed us to explore the real-world case of data with sources of imbalanced size and label distribution. A detailed discussion of the imbalanced sources is given in Appendix A.13.

Although CIFAR-10N contains real-world noise, we must still split the data into sources. For each experiment, we assign sources as to evaluate varied levels of noise and numbers of noisy sources. As is done for PTB-XL, we linearly increase the number of noisy sources from 1 to 7 (out of 10 in total), and for each set of noisy sources we linearly increase the noise level from 0.25 to 1.0.

For this, seven base model architectures are evaluated (of multiple sizes): A Multilayer Perceptron, Convolutional Neural Networks, a 1D and 2D ResNet (He et al., 2016), an LSTM (Hochreiter & Schmidhuber, 1997), and a transformer encoder (Vaswani, 2017) (Appendix A.7).

In all experiments, data points contain an observation, source, and target, which are assigned to mini-batches in the ordinary way. Features and labels are passed to the model for training, whilst sources are used by LAP (Appendix A.7). The test sets contain clean labels only.

RESULTS

Table 1: **Comparison of LAP with the baselines.** Mean ± standard deviation of the maximum test accuracy (%) of 5 repeats of the baselines and LAP on synthetic data with different noisy types. For CIFAR-100 these numbers represent the top 5 accuracy. For CIFAR-10, CIFAR-100, and F-MNIST, the number of noisy sources are 4, 2, and 6 out of 10 respectively. Unreliable sources are each 100% noisy. All values in bold are within 1 standard deviation of the maximum score. Note that IDPA and Co-teaching require almost twice the training time in comparison to Standard, ARFL, and LAP.

| | Noise Type | Model Types | | | | |
| --- | --- | --- | --- | --- | --- | --- |
| | | Standard | ARFL | IDPA | Co-teaching | LAP (Ours) |
| CIFAR-10 | Original Data | 77.91 ± 0.62 | 74.89 ± 1.67 | **79.89 ± 1.01** | **80.04 ± 0.49** | 78.34 ± 1.27 |
| | Chunk Shuffle | 71.07 ± 1.26 | 68.42 ± 0.71 | 72.68 ± 0.96 | **74.77 ± 0.3** | 73.91 ± 0.82 |
| | Random Label | 67.11 ± 1.46 | 69.93 ± 1.88 | 55.66 ± 1.54 | 68.61 ± 1.78 | **73.11 ± 0.35** |
| | Batch Label Shuffle | 68.35 ± 1.58 | 67.12 ± 1.93 | 68.19 ± 1.48 | 71.68 ± 0.92 | **73.84 ± 0.66** |
| | Batch Label Flip | 68.18 ± 1.81 | 66.16 ± 1.96 | 69.65 ± 1.54 | 71.18 ± 0.76 | **73.57 ± 0.61** |
| | Added Noise | 70.04 ± 1.09 | 68.59 ± 1.38 | 70.94 ± 1.61 | 72.29 ± 0.61 | **72.89 ± 0.37** |
| | Replace With Noise | 73.23 ± 0.7 | 67.01 ± 1.02 | 71.96 ± 1.53 | **73.87 ± 0.57** | **73.68 ± 0.4** |
| CIFAR-100 | Original Data | 76.19 ± 0.4 | 60.42 ± 1.91 | **77.78 ± 0.95** | 76.25 ± 0.54 | 75.94 ± 1.12 |
| | Chunk Shuffle | **70.23 ± 0.96** | 56.76 ± 2.69 | 66.82 ± 0.85 | **69.58 ± 0.51** | **69.98 ± 0.15** |
| | Random Label | 58.08 ± 1.22 | 48.85 ± 3.2 | 49.5 ± 1.03 | 61.2 ± 0.54 | **70.27 ± 0.85** |
| | Batch Label Shuffle | 65.23 ± 1.69 | 58.2 ± 4.45 | 64.34 ± 2.09 | **69.34 ± 0.69** | **69.58 ± 0.56** |
| | Batch Label Flip | 61.24 ± 1.32 | 56.51 ± 3.36 | 64.61 ± 2.1 | **69.04 ± 1.19** | **69.22 ± 0.45** |
| | Added Noise | 66.03 ± 0.28 | 58.35 ± 4.16 | 64.44 ± 1.11 | 66.02 ± 0.36 | **67.32 ± 0.76** |
| | Replace With Noise | **67.84 ± 1.1** | 58.88 ± 4.33 | 66.06 ± 0.42 | **68.52 ± 1.31** | **68.11 ± 1.02** |
| F-MNIST | Original Data | **83.74 ± 0.3** | 82.0 ± 0.4 | **83.69 ± 0.65** | 79.09 ± 1.14 | **83.54 ± 0.6** |
| | Chunk Shuffle | 77.4 ± 1.11 | 77.69 ± 0.51 | 77.45 ± 2.21 | 74.74 ± 1.21 | **81.74 ± 2.2** |
| | Random Label | **77.74 ± 4.3** | **77.31 ± 3.82** | **76.57 ± 7.16** | **77.41 ± 4.47** | **76.0 ± 6.24** |
| | Batch Label Shuffle | 82.25 ± 0.65 | 79.11 ± 1.71 | **82.82 ± 0.43** | 82.3 ± 0.32 | 82.01 ± 1.22 |
| | Batch Label Flip | 80.6 ± 1.73 | 78.85 ± 1.78 | **82.31 ± 0.27** | 80.43 ± 0.81 | 81.8 ± 0.71 |
| | Added Noise | 76.31 ± 2.21 | 73.28 ± 1.31 | 78.42 ± 2.1 | 75.72 ± 1.55 | **79.35 ± 0.73** |
| | Replace With Noise | 77.09 ± 2.19 | 74.12 ± 1.36 | 80.64 ± 0.42 | 78.76 ± 1.06 | **82.76 ± 0.4** |

**Baselines and LAP.** Table 1 shows the results of the baselines and LAP on CIFAR-10, CIFAR-100, and F-MNIST across varied noise. This illustrates that in a high noise setting, LAP often allows for improved test accuracy over the baselines – two different approaches to learn from noisy labels (IDPA and Co-teaching), an approach from federated learning (ARFL), as well as training a model in the standard way. Interestingly, IDPA and Co-teaching attain a higher accuracy on the original data in CIFAR-10 and 100, likely because IDPA and Co-teaching require the training of a model for twice as long or twice, *almost doubling training time* over LAP, ARFL and standard training.

On CIFAR-10 and 100, the performance improvement with LAP is most apparent with random labelling noise. For instance, on CIFAR-100, LAP achieves a roughly $+20\%$ top-5 accuracy over standard training, whilst IDPA performs worse. Although the performance improvements of LAP are less dramatic on F-MNIST due to the simpler nature of the dataset, the model still demonstrates better accuracy in certain noise conditions, particularly with chunk shuffling and replace with noise.

In these experiments, LAP either achieves the best accuracy, or is within a few percentage points of the best accuracy. In the latter case, the best accuracy is then often achieved by a method that requires longer training, but that is less robust to some noise types. Therefore, LAP on the whole achieves more consistently higher accuracy whilst requiring less compute.

To further illustrate the potential accuracy improvement from using LAP, within Appendix A.9 we also present these results as a percentage difference to the standard training method. In the following more challenging experiments, the improvement in accuracy of using LAP becomes clearer. Additionally, although IDPA and co-teaching are limited to classification tasks, LAP also works on regression (Appendix A.17).

**LAP in conjunction with RRL.** Since RRL requires significant changes to the model architecture and data augmentation, we separately test the large architecture (Appendix A.7) and experimental set-up employed in Li et al. (2021) for a fairer comparison. Here, we test RRL with and without LAP to evaluate its ability to be used in conjunction with other methods. Table 2 presents these results on CIFAR-10, and further underlines the use of LAP to improve model performance on data generated by multiple sources. Here, LAP supplements the accuracy obtained by RRL, which uses a contrastive learning approach to tackle noisy features and labels, that is improved upon by further applying our

Table 2: **RRL and RRL + LAP results.** Mean ± standard deviation of the maximum test accuracy (%) of 10 repeats of RRL and LAP on CIFAR-10 data with different types of noise. Here, 4 out of 10 sources are 100% noisy.

|  | Model Types | |
| --- | --- | --- |
| Noise Type | RRL | RRL + LAP (Ours) |
| Original Data | **87.67 ± 0.37** | **87.54 ± 0.22** |
| Chunk Shuffle | 82.93 ± 0.29 | **84.27 ± 0.31** |
| Random Label | 76.04 ± 1.43 | **80.31 ± 0.58** |
| Batch Label Shuffle | 77.66 ± 0.71 | **80.84 ± 0.51** |
| Batch Label Flip | 78.81 ± 0.66 | **82.02 ± 0.45** |
| Added Noise | 78.51 ± 0.74 | **81.70 ± 0.48** |
| Replace With Noise | **80.05 ± 0.65** | 79.00 ± 0.75 |

method to make use of the source values. We observe that LAP increases all metric scores except for "Replace With Noise", where the difference is relatively small, and "Original Data", where both methods perform equally. Again, the most substantial increase in accuracy comes from the random labelling noise, where LAP improves on RRL by around +5%. These results are supplemented by Appendix A.10, where we test different numbers of noisy sources and noise rates.

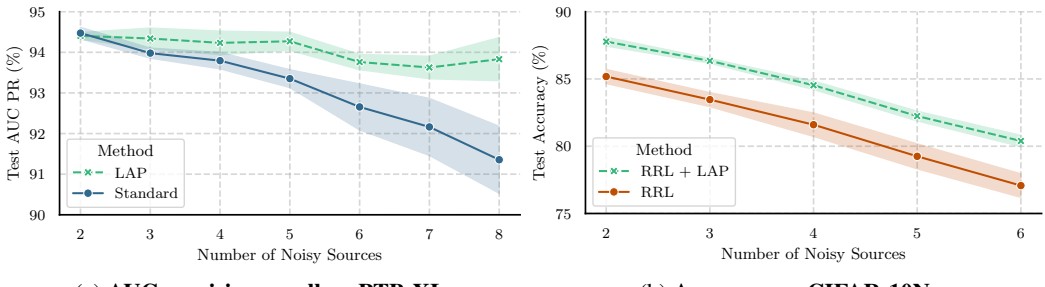

(a) **AUC precision-recall on PTB-XL.**    (b) **Accuracy on CIFAR-10N.**

Figure 3: **LAP results with a varied number of sources and noise levels.** In **(a)** we show the area under the precision-recall curve for standard training and using LAP on PTB-XL with label noise and simulated ECG interference noise for 12 total sources. In **(b)** we show the accuracy on CIFAR-10N with real human labelling noise when using RRL and RRL + LAP, with 10 sources. In both, the noise of the sources varies linearly from 25% to 100% for each number of noisy sources. The lines and error bands represent the mean and standard deviation of the maximum value for each of the 5 repeats. These figures illustrate that LAP maintains higher performance as noise rates increase.

**Varied noise level and source sizes with real-world time-series data.** Figure 3a compares the results of not using and using LAP for a time-series classification task on normal and abnormal cardiac rhythms with both label and input noise. To evaluate the model on data sources with differing noise levels, for each number of noisy sources we set the noise levels of sources at linear increments between 25% and 100%. Additionally, we linearly increase the number of these noisy sources and measure the area under the precision-recall curve (AUC PR) on the test set. These results shows that LAP allows for improved model training on data with sources of varied noise and with multiple noise types. Figure 3a also illustrates that in this setting, LAP is robust to increases in the noise within the dataset, as the AUC PR does not significantly degrade, especially when compared to not using LAP. Additionally, the standard deviation of results is smaller, suggesting LAP is more consistent in its

AUC-PR. When only two sources contain noise, the total noise rate in the dataset is only $10\%$, which has little negative effect on the standard training.

**Real-world noisy labels.** To further demonstrate the effectiveness of LAP in conjunction with other noisy data methods and on real-world noise, we now present the results on a dataset with human labelling of varied noise rates. Figure 3b illustrates the results on CIFAR-10N, a noisy human labelled version of CIFAR-10, where we observe a similar pattern to that depicted in Figure 3a. In both cases, LAP improved accuracy on noisy data at all noise levels tested, and the standard deviation over the accuracy is reduced. However, we also note that in this case the accuracy of LAP degraded faster than the AUC PR in Figure 3a as the number of noisy sources increased. This could be for various reasons, such as differing noise types, model baselines, and data type. In Appendix A.11 we also show the results of this experiment when using the same model architecture as in Table 1, and with standard training as a baseline, instead of RRL, where we arrive at the same conclusions.

Table 3: **Baselines and LAP results on a natural language task.** Mean ± standard deviation of maximum test accuracy (%) of 5 repeats of the methods on the IMDB dataset with different types of noise. Here, 4 out of 10 sources are 100% noisy.

| | Model Types | | | |
|---|---|---|---|---|
| Noise Type | Standard | IDPA | Co-teaching | LAP (Ours) |
| Original Data | 82.81 ± 0.79 | 83.24 ± 0.56 | **85.12 ± 0.8** | 83.2 ± 0.83 |
| Random Label | 65.01 ± 0.65 | 64.61 ± 0.71 | 67.18 ± 0.69 | **71.95 ± 2.94** |

**LAP on a noisy natural language task.** We also tested LAP on a natural language task, in which the goal is to predict the sentiment from the movie review. The results are presented in Table 3, for which we tested two types of noise. Here, LAP clearly out-performs the baselines by a significant margin, with IDPA not improving on standard training. Additionally, we expect that when the data is not noisy, using LAP should not limit performance, which we see here. It is interesting however, that Co-teaching performs better in the "Original Data". We hypothesise that the IMDB dataset contains some noisy labels (through human error), which were split uniformly across data sources; a limitation of our method that we discuss in Section 5.

Table 4: **Results on GoEmotions, an imbalanced sources dataset.** Mean ± standard deviation of maximum test top-5 accuracy (%) over 5 repeats of the methods on the GoEmotions data with noisy labels and random sentence permutation. Here, 30 out of 82 sources are 100% noisy.

| | Model Types | | | |
|---|---|---|---|---|
| Noise Type | Standard | IDPA | Co-teaching | LAP (Ours) |
| Original Data | **80.43 ± 0.05** | 79.66 ± 0.21 | 79.7 ± 0.31 | **80.44 ± 0.15** |
| Random Label | 76.96 ± 0.58 | 76.05 ± 0.84 | 77.05 ± 0.7 | **78.74 ± 0.41** |

**Real-world and imbalanced data sources.** To supplement the results on PTB-XL, we present LAP on an additional dataset containing real-world data sources – GoEmotions. This dataset allows us to explore the scenario in which source sizes and the classes they contain vary significantly, which could be a common scenario in real-world use cases. As LAP weights sources based on their log-likelihood, it is important to consider the robustness of our method to variations in the distribution of classes in sources, as well as their sizes. In the GoEmotions dataset, our training set contains source sizes in the range of 1 to 9320 with a mean size of 1676 and standard deviation of 1477 – providing varied source data distributions (details in Appendix A.13). Table 4 demonstrates the top-5 accuracy of the various methods in this setting, and suggests that LAP is robust to varied source data distributions in this real-world dataset. Whilst LAP and Standard training have similar performance when trained on the original data, using LAP leads to improved top-5 accuracy on the test set when random labels are introduced. On GoEmotions, random labelling had small effects on the performance of the Standard training method, reducing the test set top-5 accuracy from 80.5 to 77.0, however LAP was still able to significantly reduce that performance loss by achieving a test set top-5 accuracy of 78.7. LAP also produced greater top-5 accuracy than the baselines in all cases, and with less variability in results. In Appendix A.14 we further test imbalance in source class distributions with an extreme example.

**Additional results.** Many further experiments, such as with varied numbers of sources and source sizes (A.10 and A.12), models sizes (A.11), Imagenet (A.15 and A.16), a regression task (A.17), the effect on late training performance (A.16.1), and experiments straining the method assumptions (A.14) can be found in Appendix which provide further intuitions about LAP and strong evidence for its use in a wide variety of settings.

## 5 DISCUSSION

This research reveals some interesting future research directions. In our experiments, we study models of varied capacity (for example: Table 1 and Appendix A.11, and Table 2 and Appendix A.16), however, it is interesting to further study the effects of ill-specified models on noisy data techniques, as most methods assume that models attain smaller losses on the non-noisy data points (for example: in our work and Co-teaching (Han et al., 2018)), or that logits are reliable (for example: in RRL (Li et al., 2021) and IDPA (Wang et al., 2021)). Additionally, we focus on the setting in which knowing the source of a data point provides extra information in learning from noisy data. However, when the noise level is uniform across *all* sources, the source value does not provide additional information about a data point's likelihood of being noisy, and so in this case LAP performs as well as standard training. Here, it would be beneficial to apply a noisy data technique in addition to our method, such as with RRL + LAP, studied in the experiments (Table 2, Figure 3b, and Appendix A.10).

The additional compute required to apply LAP to a single source is $O(S + B)$ and the extra memory cost is $O(S \times H)$ (where $S$ is the number of unique sources in the dataset, $B$ is the batch size, and $H$ is the length of the loss history), as we calculate the source means and standard deviations online. For multiple sources in a batch, the additional compute becomes $O(S \times S_b + B)$ (where $S_b$ is the number of unique sources in a batch). A measured time cost for a simple experiment is given in Appendix A.18. The extra memory cost in this case is $O(S \times H + S_b \times S)$. Although vectorised in the current implementation, $O(S \times S_b)$ could be performed with $S_b$ parallel jobs of $O(S)$, since each source calculation is independent, significantly improving its speed for larger numbers of sources. However, this is already significantly faster than the baselines tested, in which Co-teaching trains two models, IDPA trains a model for twice as long, and RRL applies $k$-nearest neighbours at each epoch.

Importantly, as is the case with all methods designed for training models on noisy data, thought must be given to the underlying cause of noise. For example, in some scenarios noise within a dataset could be attributed to observations on minority groups during data collection, rather than as a result of errors in measurements, labelling, or data transfer. This is why we began with the assumption that all data (if non-noisy) is expected to be generated from the same underlying probability distribution. If data from minority groups are not carefully considered, *any* technique for learning from noisy data, or standard training could lead to an unexpected predictive bias (Mehrabi et al., 2021).

Within this work we presented LAP, a method designed for training neural networks on data generated by many data sources with unknown noise levels. In Section 4, we observed that using LAP during training improves model performance when trained on data generated by a mixture of noisy and non-noisy sources, and does not hinder performance when all data sources are free of noise. The results also illustrated that applying the proposed method is beneficial at varied noise levels, when training on multiple sources with differing levels of noise, and whilst being robust to different types of noise. We additionally see through results in Table 1, 2, 3, 4 and Figure 3a, and 3b that our method is applicable and robust across multiple domains with differing tasks. Moreover, within the Appendix we present many further experiments, showing that: (1) Our method is robust to an extreme source class distribution that tests the limits of the assumptions we made when proposing LAP (Appendix A.14); (2) The improvements in results translate to large scale datasets (Imagenet), with multiple noise types (Appendix A.16); (3) LAP is robust to overfitting of noisy data, and therefore achieves significantly greater performance during late training (Appendix A.16.1); and (4) LAP continues to outperform the baseline on a regression task (Appendix A.17). Further, our implementation (Appendix A.1) allows this method to be easily applied to any neural network training where data is generated by multiple sources and our analysis in Section 3 and Appendix A.8 provides a detailed description of the introduced parameters and their intuitions.

This work shows that LAP provides improved model performance over the baselines in a variety of noise settings and equal performance on non-noisy data, whilst being robust to a multitude of tasks and being cheaper to compute. We therefore imagine many scenarios where LAP is relevant.

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

## A APPENDIX

### A.1 CODE IMPLEMENTATION

An implementation of the proposed method, as well as the code to reproduce the results in this paper is made available. The experiments in this work were completed in Python 3.11, with all machine learning code written for Pytorch 2.1 (Paszke et al., 2017). Other requirements to run the experiments are available in the supplementary code. We make our code available under the MIT license.

All datasets tested are publicly available and easily accessible. Additionally, within the supplementary code we use the default Pytorch Dataset objects for CIFAR-10, CIFAR-100, and F-MNIST and provide Pytorch Dataset objects that will automatically download, unzip, and load the data for PTB-XL, CIFAR-10N, IMDB, and California Housing. We additionally provide code to load Tiny-Imagenet and Imagenet from a local directory, since it requires an agreement before accessing. This makes reproduction of the work presented simple to perform.

Baselines were made available by the authors on GitHub:

- ARFL (Li et al., 2020): MIT License: `https://github.com/lishenghui/arfl`.
- IDP (Wang et al., 2021): License not provided: `https://github.com/QizhouWang/instance-dependent-label-noise`.
- Co-teaching (Han et al., 2018): License not provided: `https://github.com/bhanML/Co-teaching`.
- RRL (Li et al., 2021): BSD 3-Clause License: `https://github.com/salesforce/RRL`.

All experiments were conducted a single A100 (80GB VRAM), and 32GB of RAM. The full research project required more compute than the experiments reported in the paper to design and implement LAP, as well as choose training parameters for the standard training model that were then used for LAP and the baselines. Further information about the compute time required to run each of the main experiments is found in Appendix A.7, and the corresponding section of the Appendix for further experiments. In total, using this hardware, experiments took approximately 15 days to complete.

### A.2 LOSS ASSUMPTIONS

Within this work, and for many other methods designed for tackling noisy data, we make use of the assumption that during training, neural networks learn non-noisy patterns before fitting to noisy data, and therefore achieve a greater log-likelihood on non-noisy data during early training. This is discussed within Zhang et al. (2021); Arpit et al. (2017); Han et al. (2018); Arazo et al. (2019); Yu et al. (2019); Shen & Sanghavi (2019); Li et al. (2021); Wang et al. (2021), which present both theoretical and empirical justifications. We also observe these training dynamics in Appendix A.16.1 (specifically, Figure 7), in which the standard training method overfits to noisy data points significantly during the later training stages when compared to our method for learning from noisy data.

### A.3 FURTHER DETAILS ON DESIGN DECISIONS

#### WHY USE A WEIGHTED MEAN AND STANDARD DEVIATION?

Firstly, we will discuss why the weighted mean and standard deviation is used for comparing sources with each other. This is done to allow for the identification of noisy sources by our method when different data sources might have significantly different noise levels. This is the case in Figure 2, where we have one source producing $100\%$ noise and other sources with $5\%$, and $2.5\%$ noise levels. Here, because the noise in the sources are so varied, the sources with lower noise levels would likely never have an average loss more than the unweighted mean of the loss trajectory plus the threshold created from the leniency multiplied by the unweighted standard deviation (Algorithm 1). By weighting the mean and standard deviation with the calculated temperature, we are able to ensure that all sources with noise level greater than $0\%$ will be discovered in a number of model update steps proportional to their noise level. This is shown in Figure 1, where the weighted mean and standard deviation of the loss moves further to the left, as the weight reduces the influence of the unreliable source (on the right).

WHY THE MEAN + STANDARD DEVIATION FOR THRESHOLDING?

When calculating whether a source should be considered more or less unreliable, we need to be able to calculate a reliability compared to the other sources. We therefore use a mean and standard deviation over previous loss values, so that we can calculate a *relative* reliability. As described in Section 3, it also provides an intuitive idea about the probability that a non-noisy source is incorrectly identified as noisy on a single step (if the means of loss values are assumed to follow a Gaussian distribution, which is not unreasonable according to the central limit theorem).

WHY DO WE CLIP RELIABILITY AT $0$?

In Algorithm 1, we clip $C_s$ at $0$ to ensure that when we have sources with noise levels of different magnitudes, once the sources with larger noise levels have been heavily weighted and so have insignificant effects on the weighted mean and standard deviation, we are able to start reducing the reliability of other noisy sources immediately, rather than waiting for $C_s$ to increase from some negative value to $0$. Also, since we use the $\tanh^2$ function to calculate loss (or gradient) scaling, clipping $C_s$ at $0$ prevents us from introducing an error in which loss from reliable sources get scaled with the same factor as unreliable sources.

EXPLANATION FOR USING $\tanh^2$

Equation 3 describes the scaling of gradients (or equivalently, loss values) to reduce the contributions to model updates from less reliable data sources over time. Although any monotonically increasing function can be used for $f(\cdot) : [0, +\infty) \to [1, +\infty)$, we find that $1/f(C_s) = 1 - \tanh^2(0.005 \cdot \delta \cdot C_s)$ provides some nice properties: $y = \tanh^2 x$ has gradient of $0$ at $x = 0$, and is asymptotic to $y = 1$, which are important qualities for calculating the model update scaling. The stationary point at $x = 0$ means that small perturbations in the unreliability of each source around $C_s = 0$ have little effects on the scaling of model updates, which reduces the consequences of randomness in loss values from a given source and ensures that source contributions are only significantly scaled once it is clear that their loss values are consistently larger than those of the other sources. Secondly, since $y = \tanh^2 x$ is asymptotic to $y = 1$, the scale factor, $1 - d_s$ (Equation 3) for sources that are calculated to be unreliable eventually reduces to $0$, allowing the model to "ignore" these data points as if they were masked in late training.

EXPLANATION FOR SCALING WITH $0.005$

The value of $0.005$ used within the $\tanh^2$ function allows for the depression strength parameter $\delta$ to be on the scale of $1$, as shown in Figure 2.

A.4   ALTERNATIVE INTERPRETATION

Here, we present another interpretation of our proposed method which may provide inspiration for future research.

As discussed in Appendix A.2, the model evidence, $p(\mathcal{D}|\mathcal{M})$ ($\mathcal{D}$: the dataset and $\mathcal{M}$: the model) is the probability that a dataset is generated by a given model (by marginalisation of the model parameters). In our case, this could be used to calculate the probability that data from each source, $s$ is generated by the given model, namely: $p(\mathcal{D}_s|\mathcal{M})$ where $\mathcal{D}_s$ is the data generated by a source $s$. A model designed for non-noisy data should have a high $p(\mathcal{D}_s|\mathcal{M})$ when $\mathcal{D}_s$ has a low noise level, and a low $p(\mathcal{D}_s|\mathcal{M})$ when $\mathcal{D}_s$ has a high noise level. However, $\log(p(\mathcal{D}_s|\mathcal{M}))$ is hard to compute for neural networks, and would require an approximation. It can be shown that $p(\mathcal{D}|\mathcal{M})$ can be approximated by the `logsumexp` of the log-likelihood over a training trajectory with noisy gradients, which approximates the posterior samples with SGLD or SGHMC (Chen et al., 2014; Welling & Teh, 2011). Within our work, when computing if a given source is noisy or non-noisy, the mean loss for this source is calculated over a training trajectory of length equal to the history length, $H$ (given by $\mu_s$ in Algorithm 1). This is then compared to a weighted mean of the loss over a training trajectory of length $H$ of all sources excluding the source under scrutiny (given by $\mu_{s'}$ in Algorithm 1). This weighted mean is related to $p(\mathcal{D}_R|\mathcal{M})$ where $\mathcal{D}_R$ represents the data from all non-noisy sources, since the weights allow us to reduce the influence of noisy sources. Therefore, our

method is related to one in which the model evidence is calculated for each source at each step, and where the sources with a relative model evidence higher than some threshold (defined in our work through the leniency, $\lambda$) have larger influence on the model parameters when updating the weights.

### A.5 DETAILS OF THE VARIED PARAMETER EXPERIMENTS

To build an intuition for the parameters introduced in defining LAP, we run some synthetic examples to understand their effects on $1 - d_s$ (in Equation 3). The results of this are presented in Figure 2.

**Dataset.** The synthetic data is created using Scikit-Learn's make_moons [2] function, which produced $10,000$ synthetic observations with 2 features. Each observation is assigned a label based on which "moon" the observation corresponded to. Then, to produce synthetic sources and noise levels for each source, we randomly assign each data point a source number from 0 to 4, so that we have 5 sources in total. Data points for each source were then made noisy by randomly flipping labels such that each source had a noise level of 0.0, 0.025, 0.05, 0.25, or 1.0.

**Model and training.** A simple Multilayer Perceptron (MLP) with hidden sizes of $(100, 100)$ and ReLU activation functions is trained on this data using the Adam optimiser with a learning rate of $0.01$, a weight decay of $0.0001$, and with $(\beta_1, \beta_2) = (0.9, 0.999)$ for 50 epochs with a batch size of 128. Data is shuffled at each epoch before being assigned to mini-batches.

**LAP parameters.** When conducting the experiments, the values of $\delta$, $H$, and $\lambda$ were varied as is described in the graphs shown in Figure 2 to experiment with different values of these newly introduced parameters. Unless otherwise specified, the values chosen are $H = 25$, $\delta = 1.0$, $\lambda = 1.0$.

These experiments take approximately 5 minutes to complete when using the compute described in Appendix A.1.

### A.6 FURTHER INFORMATION ON THE DATASETS

The content of each of the datasets is as follows:

- **CIFAR-10** and **CIFAR-100** (Krizhevsky, 2009) are datasets made up of $60,000$ RGB images of size $32 \times 32$ divided into 10 and 100 classes, with 6000 and 600 images per class respectively. The task in this dataset is to predict the correct class of a given image.

- **F-MNIST** (Xiao et al., 2017) is made up of 70,000 greyscale images of clothes of size $28 \times 28$ divided into 10 classes with 7000 images per class, which is flattened into observations with 784 features. The task of this dataset is to predict the correct class of a given image.

- **PTB-XL** (Wagner et al., 2020) consists of 21,837 ECG recordings of 10 second length, sampled at 100Hz, from 18,885 patients, labelled by 12 nurses. The task of this dataset is to predict whether a patient has a normal or abnormal cardiac rhythm.

- **CIFAR-10N** (Wei et al., 2022) is a dataset made up of all examples from CIFAR-10, but with human labelling categorised into different levels of quality. For our case, to allow for the most flexibility in experiment design, we utilised the worst of the human labels.

- **GoEmotions** (Demszky et al., 2020) is a natural language dataset in which the task is to correctly classify the emotion of Reddit comments from a possible 28 emotions. This dataset contains 171,820 text examples, annotated by 82 raters, with each rater contributing somewhere between 1 and 9320 labels (mean: 1676, standard deviation: 1477). To produce the training and test set, we randomly split the examples in the ratio 80:20.

- **IMDB** (Maas et al., 2011) is a natural language dataset with a sentiment analysis task. The goal is to correctly classify a movie review as either positive or negative based on its text. It contains 25,000 reviews in the training set, and 25,000 reviews in the testing set, split equally between positive and negative sentiment.

- **MNIST** (LeCun et al., 1998) contains 60,000 training and 10,000 testing images of hand-written digits in black and white at a resolution of 28. The goal of this dataset is to correctly identify the digit drawn in the image (from 0 to 9).

---

[2]Scikit-Learn:make_moons

- **Tiny-Imagenet** (Deng et al., 2009) contains 110,000 RGB images of size $64 \times 64$ from 200 classes, with the goal of correctly classifying an image into the given class. There are 100,000 training images and 10,000 images in the test set.

- **Imagenet** (Deng et al., 2009) contains 1,281,167 RGB images scaled to size $64 \times 64$ from 1000 classes, with the goal of correctly classifying an image into the given class. There are 1,231,167 training images and 50,000 images in the test set.

- **California Housing** (Pace & Barry, 1997) contains 20,640 samples of house values in California districts, with the goal to predict the median house value from the U.S census data for that region. We randomly split the data into training and testing with an $8:2$ ratio on each repeat.

**Dataset licenses.** CIFAR-10, CIFAR-100, and F-MNIST are managed under the MIT License. PTB-XL is made available with the Creative Commons Attribution 4.0 International Public License [3], CIFAR-10N is available with the Creative Commons Attribution-Non Commercial 4.0 International Public License [4], GoEmotions is made available under the Apache-2.0 license [5], and MNIST is available under the the Creative Commons Attribution-Share Alike 3.0 license [6]. IMDB[7] and California Housing[8] are made publicly available, and Tiny-Imagenet and Imagenet are available after agreeing to the terms of access [9].

**Dataset availibility.** As mentioned in Appendix A.1, all datasets tested are publicly available and easily accessible. Additionally, within the supplementary code we use the default Pytorch Dataset objects for CIFAR-10, CIFAR-100, F-MNIST, and MNIST, and provide Pytorch Dataset objects that will automatically download, unzip, and load the data for PTB-XL, CIFAR-10N, GoEmotions, IMDB, and California Housing. We additionally provide code to load Tiny-Imagenet and Imagenet from a local directory, since it requires an agreement before accessing. This makes reproduction of the work presented simple to perform.

### A.7 EXPERIMENTS IN DETAIL

For each of the experiments presented in Section 4, we will now describe the dataset, models, and training in detail.

#### MODEL ARCHITECTURES

Within our experiments, we tested many model architectures, described below. Due to limitations in compute and to allow for our work to be easily reproducible, we use different levels of model capacity which additionally illustrates that our method is applicable in varied settings. All models are also available in the supplementary code, implemented in Pytorch.

- **Low capacity multilayer perceptron (MLP):** The MLP used in Section 4 consists of 3 linear layers that map the input to dimension sizes of 16, 16, and the number of classes. In between these linear layers we apply dropout with a probability of 0.2, and a ReLU activation function. For our experiments on California Housing, we used hidden sizes of 32, 32, 32, and 1 (for the output value) with ReLU activation functions.

- **Low capacity CNN:** This model consisted of 3 convolutional blocks followed by 2 fully connected layers. Each convolutional block contained a convolutional layer with kernel size of 3 with no padding, and a stride and dilation of 1; a ReLU activation function; and a max pooling operation with kernel size of 2. The linear layers following these convolutional blocks maps the output to a feature size of 64 and then the number of classes, with a ReLU activation function in between.

---

[3]https://creativecommons.org/licenses/by/4.0/.
[4]https://creativecommons.org/licenses/by-nc/4.0/.
[5]https://github.com/google-research/google-research.
[6]https://creativecommons.org/licenses/by-sa/3.0/
[7]https://ai.stanford.edu/~amaas/data/sentiment/
[8]https://www.dcc.fc.up.pt/~ltorgo/Regression/cal_housing.html
[9]https://www.image-net.org/download

- **High capacity CNN with contrastive learning:** This CNN is inspired by the model used for the CIFAR-10 experiments in Li et al. (2021) and requires considerably more compute. It is based on the ResNet architecture (He et al., 2016), except that it uses a pre-activation version of the original ResNet block. This block consists of a batch normalisation operation, a convolutional layer (of kernel size 3, a padding of 1, and varied stride, without a bias term), another batch normalisation operation, and another convolutional layer (with the same attributes). Every alternate block contains a skip connection (after a convolutional layer, with a kernel size of 1 and a stride of 2, had been applied to the input). The input passes through a convolutional layer before the blocks. This is followed by a linear layer that maps the output of the convolutions to the number of classes. This model also contains a data reconstruction component that allows for a unsupervised training component.

- **ResNet 1D:** This model is designed for time-series classification and is based on the 2D ResNet model (He et al., 2016), except with 1D convolutions and pooling. The model is made up of 4 blocks containing two convolutional layers split by a batch normalisation operation, ReLU activation function, and a dropout layer. These convolutional blocks reduce the resolution of their input by a quarter and increase the number of channels linearly by the number of input channels in the first layer of the model. Each block contains a skip connection that is added to the output of the block before passing through a batch normalisation operation, ReLU activation function and dropout layer. This is followed by a linear layer that transforms the output from the convolutional blocks to the number of output classes.

- **ResNet 2D:** This model is designed for image classification and is exactly the ResNet 20 architecture presented in He et al. (2016) or the ResNet 18 or Resnet 50 implementation in available with Pytorch [10].

- **Transformer Encoder:** (Vaswani, 2017) This model is designed for the natural language based emotion prediction task given by the GoEmotions dataset. This model uses an embedding layer of size 256, positional encoding, and 2 transformer encoder layers (based on the implementation in Pytorch [11]) with 4 heads, and an embedding size of 256. This is followed by a linear layer that maps the output from the transformer encoder to the 28 emotion classes.

- **LSTM:** (Hochreiter & Schmidhuber, 1997) The natural language model contains an embedding layer, which maps tokens to vectors of size 256, a predefined LSTM module from Pytorch [12] with 2 layers and a hidden size of 512, a dropout layer with a probability of 0.25, and a fully connected layer mapping the output from the LSTM module to 2 classes.

DATASETS AND MODEL TRAINING

**Results in Table 1.** To produce the results in Table 1, we used two different models applied to three different datasets, with 7 noise settings.

- **CIFAR-10 and CIFAR-100:** All training data is randomly split into 10 sources, with 4 and 2 sources chosen to be noisy for CIFAR-10 and CIFAR-100 respectively. In this experiment, these noisy sources are chosen to be $100\%$ noisy so that we can get an understanding of how our method performs on data containing highly noisy sources. Noise is introduced based on the description given in Section 3. The ResNet 20 model described above is trained on this data and tasked with predicting the image class using cross entropy loss. The model is trained for 40 epochs in both cases, with the SGD optimiser and a learning rate of 0.1, momentum of 0.9, and weight decay of 0.0001, on batches of size 128. When training with LAP, we use $H = 25$, $\delta = 1.0$, and $\lambda = 0.8$ chosen using the validation data (which is made noisy using the same procedure as the training data) after trained on data with batch label flip noise. When training on CIFAR-100 we found that using a warm-up of 100 steps improved performance.

- **F-MNIST:** All training data is randomly split into 10 sources, with 6 chosen to be $100\%$ noisy. The low capacity MLP model is trained for 40 epochs using the Adam optimiser, and

---

[10] Pytorch:ResNet

[11] Pytorch:TransformerEncoderLayer

[12] Pytorch:LSTM

a learning rate of $0.001$ on batches of size $200$. When training with LAP, we use $H = 50$, $\delta = 1.0$, and $\lambda = 0.8$ chosen using the validation data (which is made noisy using the same procedure as the training data) after trained on data with batch label flip noise.

In our implementation, each data point contained an observation, the generating data source, and a target. The features and labels were passed to the model for training, and the sources were used to calculate the weighting to apply to the loss on each data point.

To allow fairness in the comparison with ARFL, global updates on this model were performed the same number of times as the number of epochs other models were trained for, with all clients being trained on data for a single epoch before each global update. All other parameters for the ARFL model are kept the same. When testing IDPA, the parameters were chosen as in Wang et al. (2021) by using the default parameters in the implementation. Finally, when testing "Co-teaching", the parameters are chosen as in Han et al. (2018) where possible, and scaled proportionally by the change in the number of epochs between their setting and our setting where they depended on the total number of epochs. The forgetting rate was set as the default given in the implementation code, $0.2$, since we do not assume access to the true noise rate.

Note that when training ARFL, since it is a federated learning method, each client is trained on sources separately.

These experiments take approximately 2 days to complete when using the compute described in Appendix A.1.

For all of the following experiments, to reduce computational cost, we fix the parameters of LAP to $H = 25$, $\delta = 1.0$, and $\lambda = 0.8$. In practise, optimising these parameters should allow for improved performance when using LAP over those presented in this work.

**Results in Table 2.** To produce the results presented here, all training data is randomly split into $10$ sources, with $4$ chosen to be $100\%$ noisy. The high capacity CNN with contrastive learning presented in Li et al. (2021) is trained for $25$ epochs, with all other parameters kept as in the original work. In addition to this model, a version (with the same parameters) is trained using LAP with $H = 25$, $\delta = 1.0$, and $\lambda = 0.8$. These models were trained on batch sizes of $128$ using stochastic gradient descent with a learning rate of $0.02$, momentum of $0.9$, and weight decay $0.0005$. Data is randomly assigned to mini-batches, with each batch containing multiple sources.

These experiments take approximately 3 days to complete when using the compute described in Appendix A.1.

**Results in Figure 3a.** This experiment allowed us to test the performance of using LAP on different numbers of sources with varied noise levels. Firstly, data is split into $12$ sources based on the clinician performing the labelling of the ECG recording. Then, for a given number of sources (increasing along the $x$ axis), the noisy sources have noise levels that are set at linear intervals between $25\%$ and $100\%$. These sources are made noisy following suggestions in Wong et al. (2012) to simulate electromagnetic interference and using label flipping to simulate human error in labelling. On this data, we train the ResNet 1D model for $40$ epochs using the Adam optimiser with a learning rate of $0.001$ and a batch size of $64$. This model is trained with and without LAP with $H = 25$, $\delta = 1.0$, and $\lambda = 0.8$. Data is randomly assigned to mini-batches, with each batch containing multiple sources.

These experiments take approximately 16 hours to complete when using the compute described in Appendix A.1.

**Results in Figure 3b.** Here, CIFAR-10 is used as the features and the labelling collected in Wei et al. (2022) are used as the noisy labels to produce CIFAR-10N. The data is first randomly split into $10$ sources. Then, for a given number of sources (increasing along the $x$ axis), the noisy sources have noise levels that are set at linear intervals between $25\%$ and $100\%$, by replacing the true CIFAR-10 labels with the real noisy labels from Wei et al. (2022). The high capacity CNN with contrastive learning presented in Li et al. (2021) is trained for $25$ epochs, with all other parameters kept as in the original work. As before, a version of this model (with the same parameters) is trained using LAP with $H = 25$, $\delta = 1.0$, and $\lambda = 0.8$. These models are trained on batch sizes of $128$ using stochastic gradient descent with a learning rate of $0.02$, momentum of $0.9$, and weight decay $0.0005$. Data is randomly assigned to mini-batches, with each batch containing multiple sources.

These experiments take approximately 30 hours to complete when using the compute described in Appendix A.1.

**Results in Table 3.** In this set of experiments, we want to test the use of LAP in a natural language setting. Firstly, we load the training and testing sets and randomly split the training set into 10 sources uniformly. We then choose 4 of the sources to be 100% unreliable, and introduce noise through random labelling and randomly permuting the order of the text. We truncate or extend each review such that it contains exactly 256 tokens. We use an LSTM to predict the sentiment of the movie reviews by training the model for 40 epochs with a batch size of 128, leaning rate of 0.001 and the Adam optimiser (Kingma & Ba, 2014). A version of this model (with the same parameters) is also trained using LAP with $H = 25$, $\delta = 1.0$, and $\lambda = 0.8$. As before, data is randomly assigned to mini-batches, with each batch containing multiple sources. Since IDPA and Co-teaching are not bench-marked on this dataset, for the baseline specific parameters we used the same values as given for CIFAR-10.

These experiments take approximately 12 hours to complete when using the compute described in Appendix A.1.

**Results in Table 4.** This experiment is designed to test the performance of LAP on a dataset with real-world data sources with significant imbalance in both their size and class distributions (further details in Appendix A.13) to better understand the robustness of our proposed method, since it calculates the log-likelihood of sources during training. We chose 30 of the total 82 raters to produce noisy labels. The Transformer Encoder was trained for 25 epochs with a batch size of 256 and a learning rate of 0.001 using the Adam optimiser (Kingma & Ba, 2014). A version of this model (with the same parameters) is also trained using LAP with $H = 25$, $\delta = 1.0$, and $\lambda = 0.8$. As before, data is randomly assigned to mini-batches, with each batch containing multiple sources.

These experiments take approximately 12 hours to complete when using the compute described in Appendix A.1.

## A.8 Assessing the sensitivity of LAP to the hyperparameters

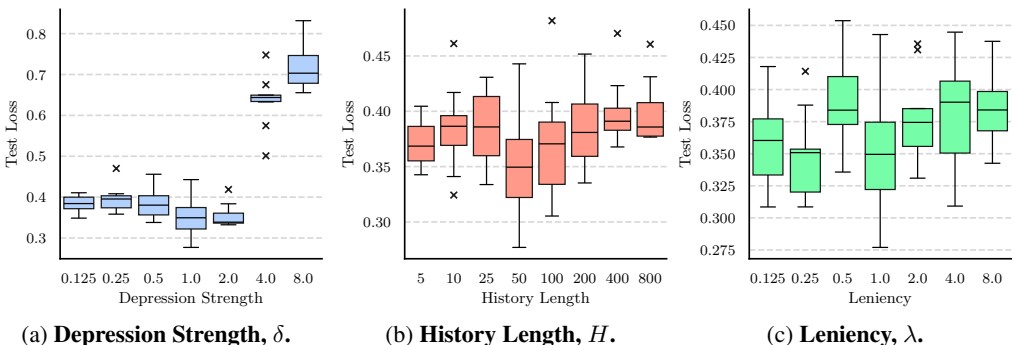

(a) **Depression Strength, $\delta$.**   (b) **History Length, $H$.**   (c) **Leniency, $\lambda$.**

Figure 4: **Sensitivity to Hyperparameters.** Here, we demonstrate the senstivitiy of the model performance based on the LAP hyperparamters introduced in Section 3 by varying their values whilst keeping the others fixed at $(\delta, H, \lambda) = (1.0, 50, 1.0)$.

In Figure 4 we present the change in cross entropy loss on the test set of the synthetic data used in Figure 2 and described in Section A.5. In this experiment the depression strength $\delta$, history length $H$, and leniency $\lambda$ were kept at $(\delta, H, \lambda) = (1.0, 50, 1.0)$ unless specified otherwise.

**Depression strength.** We observe that significantly increasing the depression strength $\delta$ can lead to a noticeable rise in test loss, particularly when $\delta \geq 2.0$ (Figure 4a). This is because large values of $\delta$ force training contributions from noisy data sources to be reduced early in training (as shown in Figure 2), whilst they might still be useful for learning a robust model. Suppressing them too quickly might prevent the model from learning important patterns in the noisy data which are useful for predicting on the test set, leading to worse performance. This highlights the need to choose a depression strength $\delta$ which strikes the right balance between filtering noise and learning from all available data. We find that a value of $\delta = 1.0$ performs well generally.

**History length.** When studying the history length $H$, we see the test loss remains relatively stable across the range of values (Figure 4b). There is some improvement for history length values of $H = 50$ and $H = 100$, but there is little difference in performance between using large and small values of $H$. This indicates that whilst increasing $H$ allows for LAP to consider more contextual information when calculating the source weighting, it also makes our method slower to react to rapid changes in the training loss due to the significantly larger history being considered. We find that a $H$ value of 25 to 50 generally performs well.

**Leniency.** In Figure 4c, we present the test loss as we vary the value of leniency $\lambda$. The loss values for this parameter remain relatively stable across its different values, suggesting the the model is fairly robust to variations in leniency. It may not be a critical parameter for tuning in this particular set up, since Figure 2 shows that for all values of leniency tested the two noisiest sources were heavily weighted during training. In scenarios where noise levels are very low, it might be necessary to reduce the leniency to capture them. In contrast, when it is thought that patterns can be learnt from the noisy data, it might be beneficial to use a larger leniency value, which intuitively lengthens the amount of time before noisy sources are weighted (Figure 2). In general, we find that a value of $\lambda = 1.0$ performs well.

These experiments take approximately 5 minutes to complete when using the compute described in Appendix A.1.

## A.9 TABLE 1 WITH PERCENTAGE DIFFERENCE IN VALUES

Table 5: **Comparison of LAP with the baselines.** Mean ± standard deviation of the percentage difference of the maximum test accuracy (%) of 5 repeats of the baselines and LAP on synthetic data with different noisy types. For CIFAR-100 these numbers represent the top 5 accuracy. For CIFAR-10, CIFAR-100, and F-MNIST, the number of noisy sources are 4, 2, and 6 out of 10 respectively. Unreliable sources are each 100% noisy. All values in bold are within 1 standard deviation of the maximum score.

| | Noise Type | Standard | ARFL | IDPA | Co-teaching | LAP (Ours) |
|---|---|---|---|---|---|---|
| | | | | Model Types | | |
| CIFAR-10 | Original Data | - | -3.88% ± 2.06 | **2.54% ± 1.59** | **2.74% ± 1.02** | 0.56% ± 1.24 |
| | Chunk Shuffle | - | -3.71% ± 1.44 | 2.28% ± 1.36 | **5.23% ± 2.10** | 4.02% ± 2.22 |
| | Random Label | - | 4.20% ± 1.76 | -17.03% ± 3.10 | 2.26% ± 3.18 | **8.98% ± 2.51** |
| | Batch Label Shuffle | - | -1.71% ± 4.75 | -0.20% ± 2.88 | 4.90% ± 2.04 | **8.07% ± 2.42** |
| | Batch Label Flip | - | -2.92% ± 3.19 | 2.24% ± 4.60 | 4.48% ± 3.84 | **7.94% ± 2.19** |
| | Added Noise | - | -2.04% ± 2.79 | 1.33% ± 3.59 | **3.24% ± 2.37** | 4.09% ± 1.40 |
| | Replace With Noise | - | -8.48% ± 2.05 | -1.71% ± 2.84 | **0.89% ± 0.80** | 0.63% ± 1.26 |
| CIFAR-100 | Original Data | - | -20.68% ± 2.62 | **2.10% ± 1.26** | 0.08% ± 0.57 | -0.32% ± 1.11 |
| | Chunk Shuffle | - | -19.13% ± 4.90 | -4.84% ± 2.08 | -0.91% ± 1.49 | -0.33% ± 1.25 |
| | Random Label | - | -15.89% ± 5.26 | -14.72% ± 3.14 | 5.40% ± 1.85 | **21.02% ± 2.47** |
| | Batch Label Shuffle | - | -10.71% ± 7.49 | -1.35% ± 2.68 | **6.38% ± 3.73** | 6.72% ± 2.69 |
| | Batch Label Flip | - | -7.78% ± 3.78 | 5.50% ± 3.09 | **12.80% ± 3.82** | 13.06% ± 2.38 |
| | Added Noise | - | -11.65% ± 5.99 | -2.39% ± 1.93 | -0.01% ± 0.89 | **1.97% ± 1.53** |
| | Replace With Noise | - | -13.17% ± 6.84 | -2.61% ± 1.76 | **1.01% ± 1.43** | 0.42% ± 2.74 |
| F-MNIST | Original Data | - | -2.08% ± 0.75 | -0.06% ± 0.55 | -5.56% ± 1.31 | -0.24% ± 1.06 |
| | Chunk Shuffle | - | 0.39% ± 1.00 | 0.08% ± 3.32 | -3.41% ± 2.60 | **5.62% ± 2.83** |
| | Random Label | - | -0.15% ± 9.40 | -1.40% ± 9.40 | -0.10% ± 9.15 | -1.87% ± 11.09 |
| | Batch Label Shuffle | - | -3.82% ± 2.22 | **0.69% ± 0.75** | **0.07% ± 1.03** | -0.29% ± 1.67 |
| | Batch Label Flip | - | -2.15% ± 2.30 | **2.17% ± 2.47** | -0.16% ± 2.95 | 1.54% ± 2.93 |
| | Added Noise | - | -3.94% ± 1.89 | **2.80% ± 2.67** | -0.75% ± 1.82 | 4.04% ± 3.08 |
| | Replace With Noise | - | -3.75% ± 4.20 | **4.68% ± 3.21** | 2.25% ± 3.88 | 7.42% ± 2.86 |

Table 5 shows the values presented in Table 1 as a percentage difference of the standard training method. This more clearly demonstrates the size of the accuracy improvement from using LAP on CIFAR-10, CIFAR-100, and F-MNIST.

In particular, using LAP leads to substantial improvements over the baseline in more challenging scenarios, such as with random labelling noise or batch label flipping. For example, on CIFAR-100 with random label noise, LAP achieves a $21.02\%$ improvement in top-5 accuracy over standard training, significantly out-performing the baselines such as IDPA. Even in cases in which LAP is not the top-performing method, its performance remains comparable and often within a few percentage points of the best results.

This illustrates LAP's robustness across varied datasets and noise types, showing that it consistently maintains or improves accuracy in high-noise conditions where other models struggle.

## A.10 RRL + LAP WITH VARIED NOISE.

Table 6: **Different noise level and number of sources.** Mean ± standard deviation (%) of the *percentage difference* in maximum test accuracy between RRL + LAP and RRL over 5 repeats when training a model on CIFAR-10 for different noise levels and numbers of sources. Here a positive value represents an improvement in accuracy when using LAP. Here, $U$ corresponds to the number of unreliable sources out of 10 sources in total. Batch label flipping was used to introduce noise.

| | Noise Level | | | |
|---|---|---|---|---|
| $U$ | 25% | 50% | 75% | 100% |
| 2 | $0.88\% \pm 0.19$ | $1.61\% \pm 0.60$ | $2.95\% \pm 0.64$ | $3.16\% \pm 0.51$ |
| 4 | $0.09\% \pm 0.71$ | $1.25\% \pm 0.58$ | $2.01\% \pm 0.96$ | $5.24\% \pm 1.24$ |
| 6 | $0.02\% \pm 0.43$ | $0.46\% \pm 0.50$ | $3.74\% \pm 0.80$ | $11.61\% \pm 1.66$ |

Table 6 shows the percentage difference in accuracy when using LAP over not using LAP for different noise levels and numbers of noisy sources.

In these experiments, data in CIFAR-10 is randomly split into 10 sources, with the row of Table 6 defining the number of noisy sources, of which all have a noise rate as given by the column of Table 6. As before, the high capacity CNN with contrastive learning presented in Li et al. (2021) is trained for 25 epochs, with all other parameters kept as in the original work. Additionally, a version of this model (with the same parameters) is trained using LAP with $H = 25$, $\delta = 1.0$, and $\lambda = 0.8$. These models are trained on batch sizes of 128 using stochastic gradient descent with a learning rate of 0.02, momentum of 0.9, and weight decay 0.0005. Data is randomly assigned to mini-batches, with each batch containing multiple sources.

When the whole dataset noise level is small (i.e: 2 sources with 25% noise), there are small performance increases when using LAP – likely because here, noisy sources have small impacts on the performance of models trained without LAP. Additionally, for lower noise levels, increasing the number of noisy sources reduced the performance improvement. This is likely because LAP is reducing noisy source training contributions early, when there is still information to learn. This can be remedied by increasing the leniency ($\lambda$), however since we were limited by compute, LAP parameters were fixed across experiments.

These experiments take approximately 5 days to complete when using the compute described in Appendix A.1.

## A.11 CIFAR-10N WITH A SMALLER NEURAL NETWORK

In Figure 5, we show the results of the experiment presented in Figure 3b except when using the Low capacity CNN described in Appendix A.7. The results in this experiment follow the same trend as in Figure 3b, reassuring us of the applicability of LAP in a variety of settings, where lower capacity models are used.

These experiments take approximately 4 hours to complete when using the compute described in Appendix A.1.

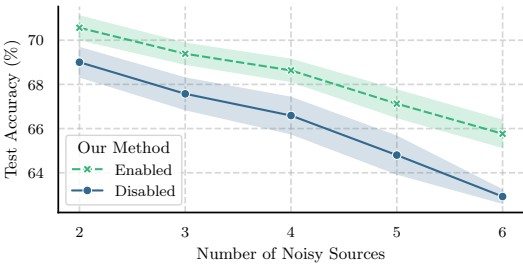

Figure 5: **Accuracy values on CIFAR-10N for an increasing number of noisy sources with a lower capacity model.** The lines and error bands represent the mean and standard deviation of the maximum test accuracy for each of the 5 repeats with random allocation of noisy sources. The noise of the sources increases linearly from 25% to 100% for each number of noisy sources. In total, there are 10 sources. Here we test the low capacity CNN (Appendix A.7).

Table 7: **Comparison of LAP with the baselines with varied numbers of sources.** Mean ± standard deviation of maximum test accuracy (%) of 5 repeats of the baselines and LAP on synthetic data with different noisy types and numbers of sources. The number of noisy sources ranges from 10 to 50,000, corresponding to "LAP-$n$", where $n$ denotes the number of sources in the training data. For all experiments, 40% of the sources were chosen as unreliable, with 100% noise rate. Since there are 50,000 training data points, LAP-50,000 corresponds to LAP acting over sources that have size 1: i.e the standard noisy data setting.

| Noise Type | Model Types | | | | | |
| --- | --- | --- | --- | --- | --- | --- |
| | Standard | IDPA | Co-teaching | LAP-10 | LAP-1250 | LAP-50,000 |
| Original Data | 69.46 ± 1.12 | **70.86 ± 0.32** | 67.73 ± 0.83 | 69.6 ± 1.08 | 69.57 ± 1.07 | 70.28 ± 0.5 |
| Chunk Shuffle | 65.92 ± 1.32 | 66.34 ± 0.99 | 63.55 ± 0.41 | **66.92 ± 0.56** | 66.08 ± 1.02 | 65.47 ± 0.98 |
| Random Label | 60.85 ± 1.67 | 57.93 ± 1.59 | 61.37 ± 1.01 | **66.5 ± 0.43** | 66.15 ± 1.16 | 66.49 ± 0.65 |
| Batch Label Shuffle | 62.23 ± 0.62 | 60.08 ± 0.13 | 63.9 ± 0.44 | 66.09 ± 0.65 | **66.71 ± 0.48** | 65.73 ± 1.13 |
| Batch Label Flip | 59.94 ± 2.48 | 61.02 ± 1.9 | 63.05 ± 2.43 | 64.48 ± 1.11 | **66.42 ± 1.22** | 65.41 ± 1.65 |
| Added Noise | 59.86 ± 1.38 | 61.19 ± 0.33 | 59.59 ± 1.37 | **63.58 ± 1.03** | 59.52 ± 1.06 | 60.55 ± 1.08 |
| Replace With Noise | **65.07 ± 0.63** | 65.85 ± 0.64 | 64.1 ± 1.13 | 64.89 ± 1.28 | **66.39 ± 1.66** | 65.09 ± 0.43 |

### A.12 CIFAR-10 WITH A WITH LARGE NUMBERS OF SOURCES

To understand the effectiveness of our method as the number of sources grows, we extended the results in Table 1 by significantly increasing the number of unique sources. Here, we use the low capacity CNN presented in Appendix A.7, with all other settings kept the same as in Table 1, except that the number of epochs is reduced from 40 to 25 as the model architecture is smaller.

In this experiment we see that LAP performs well across the different source sizes, demonstrating its robustness as the number of sources increases. In fact, LAP-50,000 corresponds to an experiment where the size of each source is equal to 1 – the standard noisy data setting. It is interesting to see that in this case LAP often performs as well or better than the baselines, suggesting its usefulness in a setting it was not originally designed for.

These experiments take approximately 12 hours to complete when using the compute described in Appendix A.1.

### A.13 SOURCE DISTRIBUTION IN GOEMOTIONS DATASET

The GoEmotions dataset (Demszky et al., 2020) was chosen for its real-world imbalanced source distributions. Here, the training set contains source sizes in the range of 1 to 9320 with a mean size of 1676 and standard deviation of 1477, enabling us to study the robustness of LAP to imbalances in the source sizes and label distributions.

In Figure 6 we show the distribution of source sizes and the number of classes within each source.

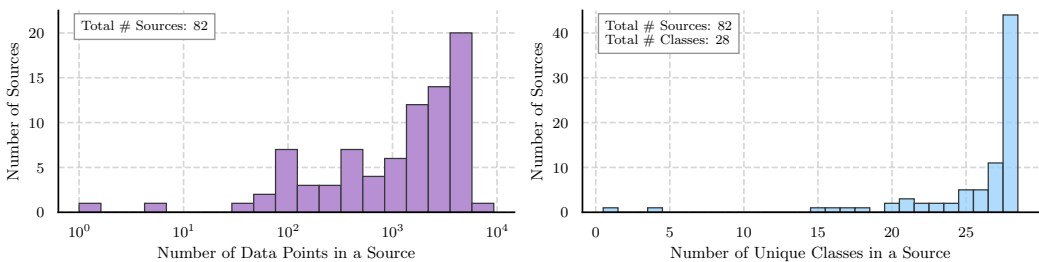

(a) **The distribution of source sizes.**    (b) **Distribution of unique classes in sources.**

Figure 6: **Source and class distributions in GoEmotions.** In **(a)** we present the number of sources with a given size, plotted on log-scale. This demonstrates the imbalance in the size of the sources in the dataset, with some containing thousands of data points, whilst others contain just a few hundred. In **(b)** we additionally explore the number of classes within each data source, showing that around half of all sources do not contain all classes, with two sources containing less than 5 classes.

Figure 6a demonstrates the imbalance in the size of each source, with some sources containing thousands of data points, whilst others contain hundreds or tens of data points. A dataset with this construction allows us to test the robustness of LAP to settings with uneven source sizes, a realistic situation in real-world data collection.

Further, Figure 6b shows the distribution of the number of unique classes that are contained within each source. In particular, we observe that around half of all sources do not contain all of the classes, with two sources containing less than 5 classes. The imbalance in the class distribution across sources is more apparent when we list the largest three classes for some of the sources:

- Source 1 with size 37: class 28 = $40.54\%$, class 19 = $8.11\%$, class 26 = $5.41\%$
- Source 2 with size 1137: class 28 = $38.43\%$, class 19 = $10.82\%$, class 2 = $5.89\%$
- Source 4 with size 2702: class 5 = $17.84\%$, class 28 = $11.84\%$, class 11 = $8.62\%$
- Source 8 with size 1253: class 28 = $14.45\%$, class 23 = $12.05\%$, class 21 = $10.45\%$
- Source 16 with size 3224: class 5 = $13.43\%$, class 23 = $12.38\%$, class 1 = $10.92\%$
- Source 32 with size 175: class 28 = $50.86\%$, class 8 = $18.86\%$, class 2 = $6.86\%$
- Source 64 with size 623: class 28 = $18.46\%$, class 4 = $11.88\%$, class 5 = $6.58\%$

These values demonstrate that sources contain large variations in the number of data points and the distributions of classes they contain. By testing with this dataset we can verify LAPs improved performance in settings in which the class distribution is uneven across sources, which may affect the source log-likelihood during training and strain the assumption we made that non-noisy sources contain similar data distributions.

### A.14 STRAINING THE ASSUMPTIONS OF OUR METHOD

To demonstrate the robustness of LAP, we now construct an experiment designed to strain the assumptions we made when introducing our method.

To work as intended, our proposed method assumes that all non-noisy sources contain similar data distributions. This allows us to say that the weighted likelihood ratio between a source under inspection and the other sources indicates whether to increase or decrease our reliability score (Equation 2). However, in real-world use cases, some sources might only produce a single class or a more challenging subset of classes. In such a case, we would like to understand whether our method can correctly identify the noisy sources without mistakenly labelling the more challenging data source as noisy.

To construct this setting, we combined data from both MNIST (LeCun et al., 1998) and CIFAR-10 Krizhevsky (2009) (Appendix A.6). First, we evenly split the MNIST data into 99 sources, and applied random label noise to 95 of them. This created a setting with many noisy sources, which

we hypothesised would be a challenging setting to learn in. We then randomly chose 2 classes from CIFAR-10 and assigned them to a single source, giving us 99 MNIST sources (with 95 of them noisy) and a single CIFAR-10 source (which remained non-noisy). We then kept the same non-noisy class labelling for MNIST from the original dataset (i.e: in the non-noisy data, a handwritten digit 0 was assigned class 0, etc) and mapped the two classes from CIFAR-10 to new classes: 10 and 11 – giving 12 classes in total. Since CIFAR-10 is more challenging to classify than MNIST, this allowed us to construct a dataset in which almost all sources are easier to classify than the final source, which contains 2 classes of more difficult data to separate. We also ran the same experiment with no random label noise applied to the MNIST sources.

To ensure the CIFAR-10 and MNIST images are the same size and have the same number of filters, we apply a grayscale and resizing transformation to the CIFAR-10 data. We then use a simple CNN consisting of two convolutional layers with 32 and 64 filters respectively, ReLU activation functions, max pooling layers (with kernel size 2), and two lineaer layers mapping representations to sizes 9216, 128, and finally 13. We use dropout, cross entropy loss, and the Adam optimiser with learning rate 0.001 and batches of size 128, trained for 25 epochs. The LAP parameters used were $(\delta, H, \lambda) = (1.0, 50, 4.0)$ for the original data experiment and $(0.1, 50, 1.0)$ for the random label experiment. These were chosen using 5 runs on a validation set.

Table 8: **Difficult data results.** Mean ± standard deviation of maximum test accuracy (%) over 5 repeats of standard training and LAP on a combination of MNIST and CIFAR-10 data with noisy labels. Here, 95 out of 100 sources are 100% noisy.

|  | Model Types | |
| --- | --- | --- |
| Noise Type | Standard | LAP (Ours) |
| Original Data | **98.38 ± 0.4** | **98.36 ± 0.39** |
| Random Label | 67.43 ± 2.83 | **96.29 ± 0.83** |

Table 8 shows the surprising results of this experiment. Since this data is made up of significant noise, it causes a large degradation in the accuracy of a standard training model when random labelling is applied to the data. However, LAP allows for almost all of this loss in accuracy to remain, showing significantly greater performance on the test set over the standard training method, whilst allowing for comparative accuracy in the absence of noise. Here, we observe that even in a contrived experiment designed as a failure case for our proposed method, LAP still produces increased accuracy over this baseline.

Table 9: **Difficult data results on the CIFAR-10 classes.** Mean ± standard deviation of maximum test accuracy (%) on the CIFAR-10 classes over 5 repeats of standard training and LAP trained on a combination of MNIST and CIFAR-10 data with noisy labels. This is the accuracy on just the CIFAR-10 test set, which is a subset of the test set corresponding to Table 8.

|  | Model Types | |
| --- | --- | --- |
| Noise Type | Standard | LAP (Ours) |
| Original Data | **0.97 ± 0.01** | 0.95 ± 0.06 |
| Random Label | **0.98 ± 0.01** | **0.97 ± 0.02** |

Additionally, Table 9 shows that we do see a small reduction in performance on the more challenging source (containing the CIFAR-10 data). On the original data, in all the five runs, LAP labelled the source containing the CIFAR-10 data as non-noisy as desired. However, when random label noise is applied to MNIST sources, the source containing CIFAR-10 data was incorrectly considered noisy in two out of the five runs. In these two cases, the validation accuracy was lower than in the other three runs:

- CIFAR-10 was incorrectly considered noisy: 23.50% and 24.39%.

- CIFAR-10 was correctly considered non-noisy: 25.46%, 25.19%, 25.53%.

Note the validation accuracy is low, because it contains the noisy labels as well as the clean labels. Therefore, with more runs and an ensemble of the higher validation accuracy runs, higher accuracy on the test set and in particular on the CIFAR-10 source, would likely be achieved.

In the extreme case presented here, which violates the assumptions we made during Section 3, we find that our method is robust to challenging source class distributions and achieves markedly greater accuracy on the test set under noisy learning, whilst achieving comparative accuracy on the single challenging (but non-noisy) CIFAR-10 source.

## A.15    TINY-IMAGENET RESULTS

Table 10: **Tiny Imagenet results.** Mean ± standard deviation of maximum test top-5 accuracy (%) over 5 repeats of standard training and LAP on Tiny Imagenet data with noisy labels. Here, 40 out of 100 sources are 100% noisy.

| | Model Types | |
| --- | --- | --- |
| Noise Type | Standard | LAP (Ours) |
| Original Data | **61.32 ± 0.61** | 60.62 ± 0.71 |
| Random Label | 46.27 ± 1.04 | **54.48 ± 0.57** |

In an effort to demonstrate the potential of LAP further, we evaluate our method on Tiny-Imagenet (Deng et al., 2009), a subset of Imagenet that contains 100,000 images from 200 classes. In this experiment, we wanted to test the use of LAP on larger images with larger numbers of sources. Firstly, we load the training and testing sets and randomly split the training set into 100 sources uniformly. We then choose 40 of the sources to be 100% unreliable, and introduce noise through random labelling. This dataset is then split in the ratio $9 : 1$ to produce a validation set. For this experiment, we train a ResNet 50 architecture using the baseline training method provided for Imagenet in Pytorch [13]. This trains the model for 90 epochs with a batch size of 256 using stochastic gradient descent with an initial learning rate of 0.1, momentum of 0.9, and weight decay of 0.0001, as well as a learning rate scheduler that multiplies the learning rate by 0.1 every 30 epochs. A version of this model (with the same parameters) is also trained using LAP with $H = 25$, $\delta = 1.0$, and $\lambda = 0.8$. As before, data is randomly assigned to mini-batches, with each batch containing multiple sources.

The results of this are available in Table 10 and again demonstrates the expected performance improvement when using LAP for datasets with sources of unknown noise, and the maintenance of performance when data is non-noisy.

These experiments take approximately 14 hours to complete when using the compute described in Appendix A.1.

## A.16    IMAGENET RESULTS WITH MULTIPLE NOISE TYPES

Table 11: **Imagenet results.** Mean ± standard deviation of maximum test top-5 accuracy (%) over 5 repeats of standard training and LAP on Imagenet data with noisy labels and noisy inputs. Here, 5 out of 10 sources are 100% noisy, with three of them containing label noise and 2 containing input noise.

| | Model Types | |
| --- | --- | --- |
| Noise Type | Standard | LAP (Ours) |
| Input and Label Noise | 68.05 ± 0.26 | **70.61 ± 0.26** |

We additionally present our results on Imagenet, which allows us to test our method on a large scale dataset. Within this experiment, as in that presented in Figure 3a, we allow for both input and label noise. In Figure 3a all noisy sources contain two types of noise, however in this experiment we allow

---

[13]https://pytorch.org/blog/how-to-train-state-of-the-art-models-using-torchvision-latest-primitives/

for different sources to contain different types of noise. We further observe, in Table 11, that using LAP improves model performance here, since it achieves a greater maximum top-5 test accuracy as expected.

For this experiment, we train a ResNet 50 architecture using the baseline training method provided for Imagenet in Pytorch[13]. This trains the model for 90 epochs with a batch size of 256 using stochastic gradient descent with an initial learning rate of 0.1, momentum of 0.9, and weight decay of 0.0001, as well as a learning rate scheduler that multiplies the learning rate by 0.1 every 30 epochs. We split the Imagenet data into 10 sources, of which 5 are chosen to be noisy (2 containing input noise and 3 containing label noise). Images are loaded as 8-bit integer arrays and interpolated to $(64, 64)$ before input noise is synthesised by randomly adding uniform integers from [-64, 64]. They are then transformed to 32-bit floats and normalised using the mean and standard deviation available on the same Pytorch training script. Label noise is added by randomly replacing labels in noisy sources. This dataset is then split in the ratio $9 : 1$ to produce a validation set for the analysis in Appendix A.16.1. A version of the ResNet 50 model (with the same parameters) is also trained using LAP with $H = 25$, $\delta = 1.0$, and $\lambda = 0.8$. As before, data is randomly assigned to mini-batches, with each batch containing multiple sources. These experiments are repeated 5 times for each setting.

These experiments take approximately 32 hours to complete when using the compute described in Appendix A.1.

### A.16.1    Late training test accuracy

Table 12: **Imagenet late training results.** Mean ± standard deviation of test top-5 accuracy (%) over the last 10 epochs of standard training and LAP on Imagenet data with noisy labels and noisy inputs, repeated 5 times. Here, 5 out of 10 sources are 100% noisy, with three of them containing label noise and 2 containing input noise.

|  | Model Types | |
| --- | --- | --- |
| Noise Type | Standard | LAP (Ours) |
| Input and Label Noise | 42.96 ± 0.64 | **66.26 ± 0.21** |

The noisy data literature often presents model performance results on the last $x$ epochs, since in reality (as we don't have access to clean test labels) there is no way of knowing when to stop training and reduce overfitting to noisy data. In our work, in an effort to be most fair to the standard training baseline, we present the maximum performance over all of training to ensure that we are not presenting results in which the standard training baseline has significantly overfit to the noisy data. However, in Table 12 we also present the average test accuracy over the last 10 epochs as is often reported in the literature. These results show the significant improvement that using LAP can have on model training when it is unknown if the standard training method is overfitting to noisy data. Similarly, Figure 7 presents the training, validation, and testing performance at each epoch during the training of a neural network using the standard method and LAP. We can see that the training curves appear usual (Figure 7a, with LAP's training curve representing the tempered loss), and that the validation accuracy is greater for standard training (since the validation set contains noisy labels), but that when tested on the clean labels (Figure 7c), it is clear that using LAP enables considerably more robustness to noisy data. Early stopping could be used here to achieve the maximally achieving models (since the epoch of maximum accuracy on the test set is the same as the validation set), which is why we find it more informative to report the maximum test accuracy as presented in all other experiments.

### A.17    California Housing: Regression results

In Table 13 we present the results of using LAP and standard training on a regression dataset in which we have 10 total sources and 4 that are 100% noisy, with labels replaced with uniform noise (sampled between the minimum and maximum label value in the training set).

To test our method against standard training, we evaluate both on a dataset of 20,640 samples of house values in California districts, with the goal to predict the median house value from the U.S

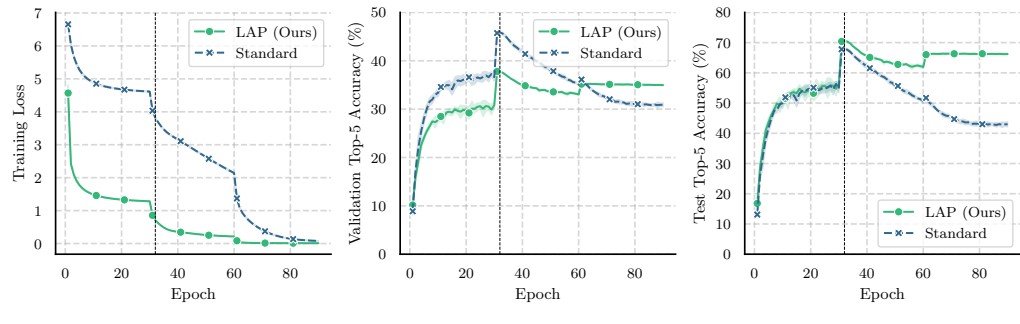

(a) **Training cross entropy loss.**  (b) **Validation top-5 accuracy.**  (c) **Test top-5 accuracy.**

Figure 7: **Performance on Imagenet.** In **(a)** we show the cross-entropy loss on the training data at each epoch where the loss using LAP represents the tempered cross-entropy loss. In **(b)** we present the validation top-5 accuracy at each epoch. In **(c)** we present the test top-5 accuracy at each epoch to demonstrate the fitting to the noise that occurs when not using LAP during late training. Here, 5 out of a total of 10 sources are 100% noisy, with 2 containing input noise, and 3 containing label noise. The two steps in performance occur at the points at which we scale the learning rate using a scheduler (epoch 30 and 60). The lines and error bands represent the mean and standard deviation over the 5 repeats. The vertical black dashed line represents the epoch at which LAP achieved maximum top-5 accuracy on the test set.

Table 13: **Standard and LAP results on a regression task.** Mean ± standard deviation of minimum mean squared error of standard training and LAP over 5 repeats on the California Housing dataset with different types of noise. Here, 4 out of 10 sources are 100% noisy. Note that "Random Label" is the only noise type with a significant difference between methods. Here, smaller is better.

| | Model Types | |
| --- | --- | --- |
| Noise Type | Standard | LAP (Ours) |
| Original Data | **0.44 ± 0.02** | **0.44 ± 0.02** |
| Random Label | 0.62 ± 0.02 | **0.45 ± 0.02** |

census data for that region – a regression task. We randomly split the data into training and testing with an $8 : 2$ ratio for each of the 5 runs we performed over each noise type. We train a multilayer perceptron as described in Section A.7 for 200 epochs with a batch size of 256, leaning rate of 0.001 using stochastic gradient descent with momentum of 0.9 and weight decay of 0.0001. A version of this model (with the same parameters) is also trained using LAP with $H = 25$, $\delta = 1.0$, and $\lambda = 0.8$. As before, data is randomly assigned to mini-batches, with each batch containing multiple sources. This was repeated 5 times for each setting.

Table 13 shows that LAP improves the test set performance significantly over the baseline for random labelling noise and maintains performance when no noise is present. This is as expected, and again illustrates its effectiveness, but this time on a regression task.

These experiments take approximately 2 hours to complete when using the compute described in Appendix A.1.

## A.18 ADDITIONAL COMPUTE EXAMPLE

In this section, we briefly describe the additional cost of applying LAP with a specific example.

In practise using our implementation, with an MLP with hidden sizes $20 - 2000 - 2000 - 5$, and batches of 512 samples (with 20 features, 5 classes, and data generated from 10 sources) the forward-backward pass through the model (without source reweighting) takes $15300 \mu$ s $\pm 164 \mu$ s and the reweighting of losses takes $469 \mu$ s $\pm 10.8 \mu$ s, increasing the time by 3%. With 50 and 100 sources, the reweighting of losses takes $1000 \mu$ s $\pm 11.3 \mu$ s and $1990 \mu$ s $\pm 49.2 \mu$ s respectively.

Using the same set-up, with batches of 2048 in size and 10 sources, the forward-backward pass (without reweighting) takes 41000 $\mu$ s ± 689 $\mu$ s whilst the loss reweighting takes 467 $\mu$ s ± 8.51 $\mu$ s (no significant change as the batch size increases), increasing the time to compute the loss and gradients on a batch by 1%.

We believe this is a fair trade-off for the potential improvement in performance.

