# OpenReview forum: "Training Neural Networks on Data Sources with Unknown Reliability"
_ICLR.cc/2025/Conference — ICLR 2025 Conference Withdrawn Submission_

### Official Review · Reviewer_8kDL · 2024-10-27

**Soundness:** 3
**Presentation:** 1
**Contribution:** 3
**Rating:** 5
**Confidence:** 4

**Summary:**

This paper addresses a novel Out-of-Distribution (OOD) learning scenario where learning occurs across multiple noisy data sources, focusing on discerning the reliability of these sources and learning the underlying iid data distribution. To this end, the Loss Adapted Plasticity (LAP) algorithm is introduced, which adaptively adjusts learning weights based on the reliability of the information. The proposed method is broadly applicable across various environments and noise types, without performance degradation in scenarios without unreliability.

**Strengths:**

1. The learning scenario proposed in this paper is of significant practical value. The modeling of data sources here shows some parallels with domain generalization, and is a useful complement to federated learning and learning from noisy data.
2. The paper provides a comprehensive empirical analysis for the proposed algorithm, validating its performance across a wide range of applications.

**Weaknesses:**

1. The main contribution of this work lies in introducing a new OOD/robust learning task. However, the problem lacks a clear mathematical formulation. While the "Motivation for New Methods" section provides an analysis and explanation of the problem, a formal proposition explicitly describing the task, learning objective, and corresponding notations would strengthen the paper.

2. The theoretical modeling of the method is somewhat simplistic. For example, in "Tempered Likelihood," data sources are modeled simply as either noisy or noiseless; however, in practice, different data sources typically contain varying degrees of noise. The mathematical model might be better represented by an IID distribution overlaid with various noise distributions of differing intensities, where noise within each data source follows the same distribution. While later experiments show that the method can indeed handle scenarios with different noise levels, the discussion in Chapter 3 is insufficiently detailed.

3. Equation 2 lacks a theoretical derivation or analysis. Under what assumptions about noise distribution was this derived, and what optimization objective led to this conclusion? Presenting it without explanation leaves it somewhat opaque to the reader. Although subsequent sentences such as "Consequently…" provide partial assumptions (e.g., NLLs), the mathematical presentation remains unclear.

4. The 0.005 value in Equation 3 is unexplained: Is it a universal value or an empirical choice for a specific task? Is this value subject to modification under different noise scenarios? While the appendix briefly discusses its role, clear criteria or rationale for selecting this value remain absent.

5. While the experimental section is extensive, the presentation is overly complex and granular, obscuring key insights. For instance, to improve clarity, a concise table summarizing key information, such as dataset descriptions, noise types, and noise intensity settings, could facilitate easier comparisons.

6. The results convincingly demonstrate the superiority of LAP over other methods, but this performance improvement seems somewhat expected given the strong alignment of the algorithm’s design with multi-source scenarios.

**Questions:**

1. Could the authors provide a formal proposition that defines the task, learning objective, and relevant notations to better clarify the problem being addressed?

2. Could the authors consider a more nuanced model of data sources that accounts for varying degrees of noise, as opposed to a binary noisy/noiseless distinction, to reflect experimental conditions more accurately?

3. What assumptions about noise distribution underlie Equation 2, and through what optimization objective was this equation derived? Could the authors provide a theoretical derivation or analysis to improve clarity?

---

### Official Review · Reviewer_ZBmo · 2024-10-31

**Soundness:** 1
**Presentation:** 2
**Contribution:** 2
**Rating:** 3
**Confidence:** 3

**Summary:**

The work attempts to address the label noise by suggesting a new algorithm for accounting for different source quality: Loss Adapted Plasticity. For this the model is pretrained on all label sources early in the training with sources deemed unreliable slowly dropping off as the time progresses.

**Strengths:**

The method shows reasonable empirical performance on constructed multi-source training problems.

**Weaknesses:**

- The writing quality can be improved. Examples of lines which I find hard to follow: 091-098;
- Notation could be made more consistent, e.g. in (1) $p(D\mid \theta)$ is used, then after (2) $p_{\theta}(D)$ is used. Do we need the $f(.)$?
- The method is basically tempering the losses from different sources to compute the total loss. Transformation in equation (1) is only loosely motivated. Do we really need to make the assumption? Also the assumption that neural networks achieve a lower empirical risk on clean data than noisy data in early training could be made more explicit in text.
- The update rule in (2) and the source reweighing in (3) miss motivation.
- The comparison to source unaware methods might not be the fairest. Federated learning and sensor fusion is invoked several times but no comparison is seen. Perhaps limiting the scope would be useful.

**Questions:**

- How do you disentangle the label noise from problem difficulty?
- Let's say we have sources arranged in a curriculum, such that once the source learns the easiest task it is much better posed to learn the more difficult one. Can the reliability estimate increase to enable learning the next source in such 'curriculum'?

---

### Official Review · Reviewer_H5eb · 2024-11-05

**Soundness:** 2
**Presentation:** 2
**Contribution:** 1
**Rating:** 3
**Confidence:** 4

**Summary:**

This paper addresses the challenge of training neural networks when data is sourced from multiple providers with varying levels of reliability. The authors propose a method called Loss Adapted Plasticity (LAP), which dynamically adjusts the training process based on the estimated reliability degree of each data source. The core idea is to proportionally reduce the influence of noisy sources during training, thereby protecting the model from overfitting to noisy data. The method is evaluated extensively across different datasets and noise types, demonstrating improved performance over standard training techniques.

**Strengths:**

1. The paper addresses the issue of how to make learning algorithms more robust when some data sources in the dataset are unreliable, which is meaningful for the application of models in real-world scenarios.

2. The experiments validate the proposed method on datasets of varying sizes, proving that the method can handle unreliable data to a certain extent.

**Weaknesses:**

1. The method proposed in this paper is too trivial. The method uses a reweighting mechanism to control the contribution of different data sources to model training, which is essentially a special case of instance reweighting. The reweighting method proposed in this paper does not offer much novelty compared to previous work, aside from being set-wise than instance-wise.

2. The core part of the proposed method is the SOURCE RELIABILITY ESTIMATION section, which directly uses the historical outputs of the model on different data sources as a basis and employs a threshold mechanism to continuously update the reliability of data sources. This approach is not elegant, for an instance, compared to existing mechanisms such as "learn to reweigh (publish in 2018)".

3. The experimental results of this paper do not show significant superiority of the proposed method over existing methods in the standard noisy label learning setting. Moreover, the experimental setup only analyzes situations where there is noise in the data, without considering more complex forms of unreliability. Additionally, the paper only considers random noise for noisy label setup and does not address more challenging scenarios like instance-dependent noise.

**Questions:**

Please refer to weaknesses.

---

### Official Review · Reviewer_i7t1 · 2024-11-11

**Soundness:** 3
**Presentation:** 2
**Contribution:** 2
**Rating:** 5
**Confidence:** 3

**Summary:**

This paper studies learning with noisy labels in a particular setup, where the data consists of multiple sources and only some sources are noisy. The authors proposed a method called Loss Adapted Plasticity (LAP), which leverages the observation that the model training loss in the early stage of training is likely a good indicator of the label noise level of a source. They validated the effectiveness of LAP on a wide variety of datasets.

**Strengths:**

* The paper is overall organized well.
* The authors conducted extensive experiments to verify the effectiveness of their method.

**Weaknesses:**

* The method needs further clarification. I am looking at Section 3, the first two subsections (Tempered likelihood and source reliability estimation). For the first subsection, the introduction of $f(C_s)$ as "temperature" is a little bit confusing here. From my understanding, temperature is usually used to scale the logits, while $f(C_s)$ here is more of like used to scale the NLL loss.

* There is a similar clarification issue for the second subsection: I wasn't able to follow the "Intuitively" part (Line 149-156). It seems to me that there's something interesting going on here when the authors are discussing the behaviors of the proposed method in cases of both when no sources are noisy and there are noisy sources. However, there is no rigorous derivation that one can follow to better understand the arguments here. It's better to formalize this part a little bit as this may be important to demonstrate the advantage of the method.

* Furthermore, the "Intuitions" subsection appears to be not necessary to me. The results here are built on synthetic toy datasets such as "make_moon", which is not that sound. To me this section is more of an ablation study of the method, studying the effect of its hyperparameters. If so, it's better to move it to the experiment section and use realistic datasets to make it more sound.

* There may be some baselines missing in the experiment section. Since the main leverage of the method is the observation that model losses in the early stage can indicate noise levels, why not design very simple baselines such as removing suspected noisy sources based on training losses? Also, it would be better to include an oracle case, for example, with noisy sources directly removed.

* Finally, I am not sure whether the studied noisy labeling case is realistic, which makes the proposed method less significant. Although the authors conducted experiments on an extensive set of datasets, all of them are at least partly hand-crafted. For example, either the noise is intentionally injected by simple label flipping (PTB-XL, GoEmotions) or the data source splits are intentionally created (CIFAR-10N). Do the authors have results on realistic data cases that are both noisy and naturally have multiple sources?

**Questions:**

Please address the weaknesses mentioned above.

---

### Note · Authors · 2024-11-21

**Comment:**

We want to thank the reviewers for taking the time to review our work. After reading the reviewers comments we have decided to withdraw our work and improve on the paper for resubmission at another venue.

**Withdrawal Confirmation:**

I have read and agree with the venue's withdrawal policy on behalf of myself and my co-authors.